



# Cosmogenic-nuclide data from Antarctic nunataks can constrain past ice sheet sensitivity to marine ice margin instabilities

Anna Ruth W. Halberstadt[1], Greg Balco[1], Hannah Buchband[1], Perry Spector[1]

[1]Berkeley Geochronology Center, Berkeley CA USA

*Correspondence to*: Anna Ruth W. Halberstadt (ahalberstadt@bgc.org)

**Abstract.** We apply geologic evidence from ice-free areas in Antarctica to evaluate model simulations of ice sheet response to warm climates. This is important because such simulations are used to predict ice sheet behaviour in future warm climates, but geologic evidence of smaller-than-present past ice sheets is buried under the present ice sheet and therefore generally unavailable for model benchmarking. We leverage an alternative accessible geologic dataset for this purpose: cosmogenic-nuclide concentrations in bedrock surfaces of interior nunataks. These data produce a frequency distribution of ice thickness over multimillion-year periods, which is also simulated by ice sheet modelling. End-member transient models parameterized with strong and weak marine ice sheet instability processes, which predict large and small sea-level impacts during warm periods, also predict contrasting and distinct frequency distributions of ice thickness. We identify regions of Antarctica where predicted frequency distributions are diagnostic of marine ice sheet instability parameterizations. We then show that a single comprehensive data set from one bedrock site in West Antarctica is sufficiently detailed to show that the data are consistent only with a weak marine ice sheet instability end-member, but other less extensive data sets are insufficient and/or ambiguous. Finally, we highlight locations where collecting additional data could constrain the amplitude of past and therefore future response to warm climates.

## 1 Introduction

The purpose of this paper is to explore how to use geologic evidence from ice-free areas in Antarctica to evaluate ice sheet model simulations of Antarctic ice sheet response to warm climates in the geologic past. This is important because ice sheet models are used to predict ice sheet (and therefore sea level) response to future climate warming, and one approach to evaluating these predictions is to compare model simulations of ice sheet change during warm periods in the geologic past with evidence for the actual ice sheet configuration during those periods (e.g., Dutton et al., 2015). The difficulty with this approach is that this evidence is (i) nearly entirely indirect, consisting mainly of proxy evidence for aggregate global sea-level change rather than direct evidence of the size or existence of a particular ice sheet or portion thereof, and (ii) often ambiguous.

As an example, we highlight the mid-Pliocene warm period (MPWP) between 3-3.3 Ma, which is proposed to be the last time that the global atmospheric $CO_2$ concentration approached the current value. Pollard et al. (2015) carried out MPWP Antarctic





ice sheet model simulations and showed that model sea level contributions from the Antarctic ice sheet were strongly dependent
on the model treatment of nonlinear feedback processes active at marine ice margins (Fig. 1). Model runs with strong marine
ice margin instabilities simulate complete deglaciation of both the central West Antarctic Ice Sheet (WAIS) and marine basins
around the East Antarctic margin, with a global sea level contribution up to 17 m (Fig. 1b). Runs lacking these instabilities
trigger deglaciation of the WAIS but not East Antarctic basins, limiting the sea-level contribution to 2-4 m (Fig. 1c). Clearly,
these end members imply significantly different potential sea-level contributions from Antarctica during future climate
warming. However, far-field sea-level data for the MPWP have been interpreted to be consistent with both simulations
(Winnick and Caves, 2015; Rovere et al., 2014; Balco, 2015) and, so far, do not provide strong evidence in favour of one or
the other.

The aim of this paper is to explore how to use geologic data from the Antarctic continent to differentiate between ice sheet
model simulations with stronger and weaker marine ice margin instabilities (e.g., Fig. 1b versus 1c), thus providing insight
into which model more accurately represents the true ice sheet response to warm climates. We describe these end-member
simulations as 'sensitized' or 'desensitized' models based on the idea that stronger positive feedbacks in the form of marine
ice instabilities result in model predictions that are more nonlinear, that is, more "sensitive," with respect to the forcing.
Specifically, we investigate the sensitivity of ice sheets to (a) ocean temperatures and (b) hydrofracture of ice shelves. Our
basic chain of reasoning in exploring how to differentiate between these two model end members is as follows.

• The critical difference between sensitized ice sheet models (with strong marine ice margin instabilities) and
desensitized models (with weak instabilities) is the extent of deglaciation of marine basins. Because deglaciation of
marine basins leads to larger sea-level impacts, it is also the element of model prediction that is most of concern in
future scenarios.

• Ideally, the best way to test a sensitized model that predicts large-scale deglaciation of marine basins in past warm
climates would be to obtain geologic evidence from beneath the present ice sheet in these basins that could show
unambiguously whether the basins had, in fact, deglaciated. Unfortunately, although subglacial access drilling is
under development, this is not possible at the moment.

• Although the differences between desensitized and sensitized model behaviour are most important for areas that are
currently ice-covered, the models also make predictions about the ice cover history of areas that are currently ice-
free. In contrast to subglacial basins, it is possible to gather geologic data from ice-free outcrops.

• Therefore, our goal is to quantify if, where, and when sensitized and desensitized model simulations make different
ice cover history predictions for Antarctic outcrops where corresponding geologic data already exist or could be
collected. At these locations, we can compare model predictions to geologic data as a means of gaining insight into
whether sensitized or desensitized models are more accurate representations of ice sheet behaviour.






Specifically, we target bedrock surfaces that are repeatedly covered and uncovered by ice as the ice sheet expands and contracts during glacial-interglacial cycles. The accumulation of cosmogenic nuclides during cycles of exposure provides a geologic measurement of integrated ice cover frequency over long periods of time (Section 2). Long-term transient ice sheet models predict the same quantity – the frequency distribution of ice thickness at some location in the ice sheet. We describe sensitized

and desensitized ice sheet model experiments (Section 3) and show that these simulations predict distinct and contrasting frequency distributions over parts of the ice sheet (Sections 4 and 5). At suitable bedrock outcrops where model predictions diverge (Sections 6.1 and 6.2), geologic data from a single location can be used to constrain the fundamental behaviour of the entire ice sheet. We benchmark our model simulations with existing geologic data (Sections 6.3 and 6.4) and make recommendations for future targeted sampling to further elucidate past ice sheet sensitivity to marine ice margin instabilities

(Section 6.5).

## 2 Geologic reconstructions of long-term ice cover frequency

In the interior of Antarctica, bedrock surfaces of mountain peaks that protrude above the ice sheet as nunataks have been shown in many studies to have extremely high concentrations, higher than anywhere else on Earth, of cosmic-ray-produced nuclides that are used to quantify durations of surface exposure (Nishiizumi et al., 1991; Brook et al., 1995; Ivy-Ochs et al., 1995;

Bruno et al., 1997; Schafer et al., 1999; Margerison et al., 2005; Mukhopadhyay et al., 2012; Spector et al., 2020). These observations show that these bedrock surfaces have been exposed to the cosmic-ray flux at the Earth's surface without appreciable weathering or erosion for, in many cases, millions of years.

Glacial-geologic observations and cosmogenic nuclide measurements have also demonstrated that many such bedrock surfaces have been repeatedly covered by the Antarctic ice sheet in the past. This cosmogenic-nuclide evidence consists of

measurements of the ratios of cosmic-ray-produced radionuclides with different half-lives: while the absolute concentration of a cosmogenic nuclide reflects the integrated duration of surface exposure, the ratio of two nuclides reflects whether or not this exposure was continuous or interrupted by periods of cosmic-ray shielding under an expanded ice sheet (Dunai, 2010). The existence of surfaces with very old total exposure ages despite repeated glaciation is possible because past ice cover has been frozen at the bed and therefore non-erosive. Bedrock surfaces are essentially unmodified during periods of ice cover, and

during ice-free periods they continue to accumulate additional cosmic-ray dose.

As described by Spector et al. (2020) and also Balco et al. (2014), this general principle can be applied to interpret measurements of multiple cosmogenic nuclides in bedrock surfaces as a quantitative estimate of the average fraction of the time that the bedrock surface has been covered by ice during its recorded exposure history. In brief, the fraction of the time that the surface is ice covered, as mentioned above, is related to the ratios of multiple cosmogenic nuclides. The length of the

recorded exposure history is inferred from the nuclide concentrations (higher nuclide concentrations indicate a longer total exposure history) and also the half-lives of the measured radionuclides (a short half-life nuclide "forgets" information about





events older than several half-lives), and is commonly as long as several million years at interior Antarctic sites. Thus, multiple-nuclide data from a single bedrock sample record the average ice cover frequency at the sample site over a long period of time.

The method for inverting cosmogenic-nuclide data for ice cover frequency involves several additional assumptions, mainly
having to do with whether bedrock surface erosion is steady or episodic, and an algorithm for testing these assumptions using the relationship of data from adjacent elevations, all of which are described in detail in Spector et al. (2020). In general, a data set that has more samples, spans a larger elevation range, has samples more closely spaced in elevation, and includes more different nuclides provides more opportunities for internal validation and therefore a higher-confidence reconstruction of ice cover frequency. Although the assumptions can be (and should be) questioned for some field situations and data sets, the
purpose of the present paper is to explore how ice cover frequency reconstructed from geologic data can be used to test model simulations. Thus, to proceed, we accept that these reconstructions are accurate and have not included a detailed assessment or justification of this assertion. Information needed for a more comprehensive assessment of the approach can be found in Spector et al. (2020) and Balco et al. (2014).

If multiple-nuclide data from a single sample provide the ice cover frequency at one sample site, data collected from multiple
bedrock samples spanning a range of elevations therefore provides the average ice cover frequency at a range of elevations. The ice cover frequency at a range of elevations, in turn, is equivalent to the cumulative frequency distribution of ice thickness at the location of the samples. We focus on this quantity – the cumulative frequency distribution of ice thickness, or "ice thickness CDF" – because it is important for two reasons. First, the ice thickness CDF is also a prediction derived from long-term transient ice sheet modelling, which provides the opportunity to directly compare model predictions with geologic
observations. Second, the ice thickness CDF is diagnostic of the degree of ice sheet model nonlinearity. Thus, comparison of reconstructed and modelled ice thickness distributions is a potential means of using geologic data that exist now or can be easily gathered in the future to test whether nonlinear ice sheet models that predict catastrophic sea-level impacts in future-analogue climates are or are not an accurate representation of past ice sheet change.

## 3 'Sensitized' versus 'desensitized' ice sheet modelling

Previous ice sheet modelling has shown that strong modelled ice margin feedback processes trigger the complete deglaciation of marine-based ice, whereas model runs with less sensitive parameterizations produce more limited sea level contributions from the Antarctic Ice Sheet under warmer-than-present climates (Pollard et al., 2015; DeConto et al., 2021). In this work, we produce two end-member ice sheet model simulations that are either strongly or weakly sensitive to marine ice instabilities and characterize the differences in ice sheet evolution between the two end-member scenarios. Geologic records of long-term
cosmogenic exposure histories across the Antarctic continent can then be used to test which of these modelled ice thickness pattern better represents past ice sheet behaviour, thereby shedding light on the Plio-Pleistocene sensitivity of the Antarctic Ice Sheet to marine feedbacks and instabilities.



We describe our two end-member ice sheet model simulations as 'sensitized' or 'desensitized', referring to their parameterized

sensitivity to marine ice sheet feedbacks. This concept is similar to the heuristic description of ice sheet behaviour in some paleoclimate literature as 'dynamic' or 'stable', based on the tendency of the ice sheet to experience large and/or rapid variations in total ice volume (e.g., Sugden et al., 1993; Bart and Anderson, 2000; Naish et al., 2009; Levy et al., 2016). In this conceptualization, the sensitized ice sheet produces a stronger nonlinear response to a forcing (due to enhanced sensitivity to marine ice margin instability feedbacks). That is, given the same external temperature and accumulation forcing, the sensitized

model will gain and lose ice faster and to a greater extent than the desensitized model because these positive feedback processes cause the system to shift more quickly between equilibrium states. This produces rapid rates of change in between maximum and minimum ice sheet configurations. Strong positive feedback mechanisms also drive more extreme maximum and minimum ice sheet configurations because they trigger runaway feedbacks that proceed in the absence of additional forcing to grow or shrink the ice sheet. Although all ice sheets experience both linear and nonlinear processes, the desensitized model is

characterized by more linear behaviour (incremental forcing produces a constant proportionate ice mass loss or gain). The desensitized parameterizations make this ice sheet model endmember less sensitive to nonlinear instability feedback mechanisms.

Sensitized and desensitized behaviour is characterized using a conceptual example in Figure 2. For a given forcing (black bars, Fig. 2a), the desensitized model (blue line) produces ice volume fluctuations that are proportional to the forcing (Fig. 2d).

Because the desensitized sheet generally responds linearly to the given climatic forcing, it spends more time in an intermediate configuration, so ice volume, like the state of the forcing function, is normally distributed (Fig. 2b,c, blue). The sensitized model (dashed red line, Fig. 2a) responds nonlinearly to the forcing (due to strong positive feedback mechanisms; Fig. 2e), and produces a bimodal frequency distribution (Fig. 2b,c, red) that reflects the tendency to occupy extreme minimum or maximum states.

In this work, we run two end-member (sensitized and desensitized) ice sheet simulations transiently across the last 5 million years. In these simulations, end-member parameter choices influence the modelled ice sheet sensitivity to marine ice margin feedbacks. Specifically, we tune parameterizations related to (a) ocean temperature fluctuations across a glacial cycle, and (b) ice cliff hydrofracture instability.

## 3.1 Parameterized marine ice margin instabilities

Although elevated ocean temperatures are not an instability mechanism by themselves, warm (subsurface) ocean temperatures can erode marine grounding lines and trigger marine ice sheet instability on reverse-sloping beds (Schoof, 2007; Pritchard et al., 2012; Favier et al., 2014; Smith et al., 2020). Ocean melt at the base of ice shelves also accelerates ice mass loss: as ice shelves thin and disappear, the buttressing force (backstress) that holds back upland grounded ice is reduced, causing glacier



velocities to increase and discharge more ice into the ocean (Reese et al., 2017; Gudmundsson et al., 2019). In addition to
ocean melt-driven feedbacks, ice shelves are susceptible to surface melt processes that drive hydrofracture. Liquid meltwater
forming on top of ice shelves can exploit existing crevasses, further propagating crevasse penetration until fracture occurs
through the full thickness of the shelf (e.g., Scambos et al., 2003; Nick et al., 2010). In places where thick grounded ice reaches
the ocean, this process exposes very tall ice cliffs which are structurally unstable and fail under their own weight in a positive
feedback loop that drives 'marine ice cliff instability', e.g., ice sheet collapse in deep marine basins (such as the WAIS and
portions of the EAIS; Pollard et al., 2015; DeConto and Pollard, 2016).

Both of these ocean-melt and ice-cliff-hydrofracture mechanisms can trigger rapid and non-linear retreat of the marine ice
sheet margin once a climatic threshold is attained (approximately +2-3°C ocean warming has been estimated to drive WAIS
collapse, for example, in Sutter et al., 2016; and surface melt rates of 750 mm/yr is thought to produce enough liquid melt
water to activate marine ice cliff instability in places with thick marine grounded ice and pre-existing surface crevasses, e.g.,
Trusel et al., 2015, Pollard et al., 2015). In our two end-member simulations, parameter values influence the threshold for non-
linear ice sheet response to a climatic forcing; both the sensitized and desensitized simulations exhibit some non-linear
behaviour, but when parameter values are high (e.g., in the sensitized model), thresholds are exceeded more often and non-
linear behaviour dominates (see Fig. 2).

We use an established ice sheet/shelf model (DeConto et al., 2021) with hybrid dynamics (Pollard and DeConto, 2012) to run
transient simulations across the last 5 Myr. This computational effort requires a relatively coarse grid resolution (40km,
although model behaviour is fairly insensitive to grid size; ref) as well as highly parameterized surface and ocean temperature
forcings. We therefore use a climate weighting scheme following the approach of Pollard and DeConto (2009): modern input
climate forcing datasets (surface air temperature, precipitation, and subsurface ocean temperatures) are scaled based on a
combination of factors (Antarctic summer insolation; benthic $\delta_{18}O$; and atmospheric $CO_2$ concentration). We implement an
additional ocean temperature parameterization that determines the amplitude of this scaling by specifying the maximum and
minimum uniform temperature shifts that are applied within the weighting scheme. In other words, for time periods when the
computed climate 'weight' is at a minimum (e.g., insolation parameters, oxygen isotope values, and $CO_2$ concentrations similar
to the Last Glacial Maximum), the modern ocean temperature field is uniformly lowered by the specified amount. As climatic
conditions approach modern values, the ocean temperature shift correspondingly approaches zero. As the climate warms above
modern, a positive ocean temperature shift is applied.

For the sensitized model simulation, the uniform ocean temperature shifts range from +3 to -3°C. These values reflect our
estimates of the most extreme temperature shifts that could have reasonably occurred during glacial and interglacial periods,
guided by the existing literature. For example, Dowsett et al. (2009) reconstruct global Pliocene surface ocean temperatures of
about 2°C warmer than today, and DeConto & Pollard (2016) simulate Pliocene conditions by adding a uniform +2°C
temperature shift to their modelled oceans. We also query a coupled atmosphere/ocean model simulation of the last



deglaciation (TraCE-21k; Liu et al. (2009) which simulates subsurface (400m) ocean temperatures 2-3°C cooler around Antarctica at the last glacial maximum. In the sensitized model simulation, ocean temperature shifts range from +1 to -1°C, representing our conceptualization of an ice sheet system where ocean temperatures less frequently trigger non-linear feedbacks of ice growth and decay. Both the sensitized and desensitized ocean temperature scaling parameterizations yield

reasonable glacial maximum extents at ~20-15 ka with subsequent retreat to approximately modern configurations by 0 ka.

The sensitized model simulation also includes an additional ocean warming factor of (1.5°C) applied only to the Amundsen Sea region (cf. DeConto & Pollard, 2016; DeConto et al., 2021); this correction reflects the recent subsurface ocean warming in this area. Without further information about past time periods, for the sensitized simulation, we assume that this recent warming trend is a signature of warmer intervals and therefore apply it during interglacials. For the desensitized simulation,

we further suppress nonlinear response to climate warming by assuming that this recent warming trend is simply "noise" and do not apply it in past warm interglacials.

A key non-linear feedback process governing ice-sheet behaviour during warm worlds is the hydrofracture of ice shelves and subsequent marine ice cliff instability (Pollard et al., 2015) that triggers ice-sheet collapse. Two parameterizations govern the modelled ice-sheet sensitivity to marine ice cliff instability. A crevasse propagation parameter ('CALVLIQ'; see DeConto et

al., 2021) dictates how much existing crevasses will deepen in response to the accumulation of liquid water on the ice surface, e.g., how sensitive ice shelves are to crevasse penetration which causes ice shelf collapse via hydrofracture. A cliff collapse 'speed limit' parameter ('VCLIFF'; DeConto et al., 2021) sets the maximum rate of horizontal ice cliff wastage once the ice shelf is gone. The sensitized model simulation uses the largest values considered in DeConto et al., 2021 (CALVLIQ=195 m/(m/yr)$^2$ and VCLIFF=13 km/yr). This maximum VCLIFF value of 13 km/yr is based on observed velocities at Jakobshavn

Isbrae (Joughin et al., 2012; Pollard et al., 2015). In the desensitized model, both parameters are set to 0, effectively turning off the marine ice cliff instability feedback. While brittle fracture and crevassing can still occur, the additional liquid water accumulation does not further propagate crevasse penetration, and ice cliffs cannot retreat even when they would theoretically fail.

In this work we focus on marine ice instability mechanisms; although surface mass balance feedbacks may introduce some

nonlinear behaviour (e.g., Weertman, 1961), both end-member simulations should be impacted equally.

**4 Modelled Antarctic ice sheet thickness in long-term, transient end-member simulations**

Model simulations of Antarctic ice sheet evolution spanning the past 5 Myr produce different characteristic patterns of ice-sheet behaviour, depending on the parameterized ice sheet sensitivity to marine feedbacks and instabilities (Fig. 3). The sensitized end-member model (with parameter values that enhance ice sheet sensitivity to marine ice margin instability

feedbacks) produces more non-linear behaviour, with more extreme minimum and maximum ice sheet configurations, more time spent in these fringe configurations, and rapid rates of change between these states. Conversely, the desensitized model



is characterized by more linear behaviour, with more time spent in intermediate configurations. This is reflected in a histogram of ice volume (Fig. 3b) showing that the desensitized ice sheet is normally distributed (more frequently has an intermediate value) whereas the sensitized ice sheet is bimodally distributed (more frequently occupies extreme maximum or minimum

configurations). The frequency distributions of sensitized and desensitized simulations are distinct from the model forcing time series (LR04 $\delta_{18}$O stack; Fig. 3c), indicating that our selected model parameterizations (rather than the properties of the forcing dataset) are the primary control on characteristic model behaviour.

Our simulations of sensitized and desensitized ice sheet behaviour closely resemble the conceptual example in Section 3, but use a robust numerical model with realistic physics (Fig. 3a,b) rather than a sample dataset that was 'non-linearized' using a

simple exponential transformation (Fig. 2a,b). This confirms that our model approach has successfully promoted 'linear' vs.'non-linear' ice-sheet behaviour by varying the parameterized ice sheet sensitivity to marine ice feedbacks. These end-member simulations produce contrasting patterns of ice sheet fluctuation that leave inherently different characteristic imprints on the geologic record.

## 5 Computing ice thickness frequency distribution as a metric for model/data comparison

This section describes the metric that we use to identify differences between end-member ice sheet model predictions for comparison with geologic observations. As described in Section 2, we focus on cumulative frequency distributions for model ice thickness – the 'ice thickness CDF' – because this is equivalent to the cumulative ice cover frequency that is inferred from bedrock cosmogenic-nuclide concentrations at interior nunataks. The ice thickness CDF is therefore both a geological observable and a model prediction.

### 5.1 Modelled ice thickness frequency distributions at a discrete location

The ice thickness CDF at many locations in Antarctica differs between sensitized and desensitized model runs in the same way as ice volume: the sensitized model tends to spend more time at extreme values of ice thickness and less time at intermediate values. Figure 4 shows an example for a nunatak in the interior of the West Antarctic Ice Sheet: although the total range of ice thickness at this site is nearly identical for both models, the sensitized model is more likely to occupy minimum (ca. 1000 m

for this example) or maximum (ca. 1500 m) values, whereas the desensitized model is more likely to occupy intermediate values near 1200 m. Fig. 4(d) shows the currently exposed nunatak on the same elevation axis as modelled ice thickness patterns at this site; as the sensitized and desensitize models simulate glacial/interglacial ice thickness fluctuations, the nunatak is periodically covered and uncovered (at this particular site, the top of the peak is never ice-covered above ca. 1500 m).



### 5.2 Computing the difference metric between modelled ice thickness frequency distributions

First, we aim to identify regions of Antarctica where the difference between ice thickness CDFs simulated by end-member models is as large as possible, and therefore might be easiest to distinguish using geologic data. To accomplish this we use a simple difference metric, henceforth the 'CDF difference metric', defined as follows:

Given two ice thickness CDFs, we define an evenly spaced mesh of cumulative frequency values $f_i$ = {0.1, 0.2, 0.3,…,0.9} and identify the corresponding elevations $h_i$ in each model distribution (Figure 5). Given sets of such elevations for desensitized

($h_{d,i}$) and sensitized ($h_{s,i}$) models, we define the CDF difference metric $D$ to be the sum of the squared differences at the mesh points:

$$D = \sum_i (h_{d,i} - h_{s,i})^2 \qquad\qquad (1)$$

The omission of the end-member frequencies (0 and 1) from $f_i$ suppresses pathological results that can be caused by a few extreme values at the ends of the ice thickness distribution. As shown in Figure 5, this metric highlights differences between

unimodal and bimodal thickness distributions characteristic of the desensitized and sensitized model runs.

### 5.3 Spatial patterns in the difference metric between modelled ice thickness frequency distributions

Here we compute the CDF difference metric 'D' for every grid cell across the Antarctic model domain. The resulting map (Fig. 6) reveals that the sensitized and desensitized end-member simulations are generally most similar in the EAIS interior (lighter reds) and most different across areas of marine-based ice (darker reds). This pattern reflects the propensity of the

sensitized run to simulate a fully grounded or fully collapsed ice sheet (i.e., produce a bimodal ice thickness CDF) in places where marine margins are susceptible to hydrofracture and ocean temperature feedbacks: the WAIS, EAIS marine basins (for example, Wilkes Subglacial Basin, Fig. 6a), and around the currently deglaciated continental shelf where expanded ice sheets would have been grounded below sea level. In contrast, the desensitized ice sheet advanced and retreated across these marine regions more slowly and linearly (e.g., Fig. 2f vs 2g). In the central EAIS and ice divide areas, ice thickness patterns are more

sensitive to interior accumulation rates rather than dynamic thinning induced by changes near the grounding line, and therefore vary little between models (low values of ln(D) in Fig. 6).

Our CDF difference metric varies slightly depending on the time period considered. Fig. 6a shows values of D across the full 5 Ma extent of the simulations, whereas Fig. 6b and 7c consider only more recent time periods (the Pleistocene, 2.6 Ma - present, and post-Mid-Pleistocene-Transition, 1.2 Ma - present, respectively). Cosmogenic nuclides have different half-lives,

and therefore ice thickness CDFs from geologic data should be compared with model results integrated across the same time period as the data. For example, one commonly measured cosmogenic nuclide in Antarctic bedrock surfaces is aluminum-26, which has a half-life of 0.7 Ma. Therefore, an integrated ice cover history based on [26]Al measurements will be biased towards



events in the past 2-3 half-lives, or ~1-1.5 Ma. $^{26}$Al produced more than 4-5 half-lives ago will no longer be detectable at all, so $^{26}$Al data can provide no information about events prior to ~3 Ma. The other most commonly measured nuclides are beryllium-10, which has a half-life of 1.4 Ma, and neon-21, which is stable. $^{10}$Be concentrations therefore provide information primarily about events in the last ~3-4 Ma, and $^{21}$Ne concentrations, theoretically, back to the original formation age of rock surfaces.

The time period of integration is important for some regions of the ice sheet. For example, in the region of the Wilkes Subglacial Basin (grey box, Fig. 6a), the CDF difference metric is much higher for the full 5 Ma Plio-Pleistocene model run than for the post-2.6 Ma and post 1.2 Ma periods. The generally smaller Pliocene ice sheet provided more opportunities for marine ice margin retreat and basin deglaciation and therefore more opportunities for sensitized and desensitized models to exhibit divergent behaviour. Larger and more extensive ice sheets in the later Pleistocene provide fewer such opportunities. The importance of this is that for this region, ice cover frequency estimates based on longer-half-life cosmogenic-nuclides (e.g., $^{10}$Be and $^{21}$Ne) would potentially allow model end members to be distinguished, but estimates based on shorter-half-life nuclides ($^{26}$Al) would not.

## 6 Discussion

Here we describe the specific ice thickness CDF and bedrock outcrop characteristics (Sections 6.1 and 6.2, respectively) that make a bedrock site potentially suitable for testing Antarctic ice sheet sensitivity to nonlinear marine ice margin instabilities. We outline five criteria to identify locations where long-term cosmogenic nuclide data could be used for such model/data comparison. We then proceed to benchmark end-member model simulations using sites where ice-cover frequency data currently exist (Sections 6.3 and 6.4), and consider target locations where future field expeditions could potentially collect additional data to build a more robust understanding of past ice sheet behaviour (Section 6.5).

### 6.1 Site selection criteria based on characteristic ice sheet behaviour

We have previously demonstrated that ice sheet model simulations with stronger and weaker parameterizations of marine ice sheet feedbacks produce divergent ice sheet behaviour across millions of years, and that these end-member models produce contrasting and distinct ice thickness CDFs in some regions but not others. Figure 7 compares CDF difference profiles for the sensitized and desensitized simulations at some discrete locations around Antarctica where bedrock surfaces are known to record multi-million-year exposure histories. This reveals several situations where model benchmarking could or could not be possible.

Guided by the CDF profiles highlighted in Figure 7, we identify the specific properties of modelled ice thickness CDFs that characterize suitable locations for long-term model data comparison.



*Criterion 1: Ice thickness CDFs diverge between sensitized and desensitized ice sheet models at bedrock outcrop locations.* The behaviour of sensitized and desensitized ice sheet models must be sufficiently different (e.g., the CDF difference metric D must be large) to be able to use cumulative ice frequency data to distinguish between model predictions. Locations are

unsuitable for this purpose if the ice thickness CDFs are similar. For example, at the Grove Mountains in East Antarctica (Fig. 7a), sensitized and desensitized models predict nearly indistinguishable ice thickness CDFs, so this site would not be useful for differentiating between models.

On the other hand, there exist many locations where model CDFs are distinct throughout their elevation range and where nearby geologic data could be collected. For example, the Pirrit Hills in West Antarctica (Fig. 7e, Fig. 4), display significantly

different ice thickness CDFs, and, as discussed below in Section 6.3, extensive cosmogenic-nuclide data have been collected from Pirrit Hills sites and indicate multimillion-year exposure histories for bedrock surfaces. Other examples where data-model comparison could be possible based on this criterion are near major outlet glaciers such as the Lambert Glacier (Fig. 7c,d), the Recovery and Slessor Glaciers in the Shackleton Range (panels g and h), and the Beardmore (i,j) and Byrd (k,l) Glaciers in the Transantarctic Mountains. All these glaciers are close to numerous ice-free bedrock outcrops where geologic

data either have been or could be collected.

Generally, the largest values of D occur mostly in subglacial basins and coastal areas (because these regions are more vulnerable to marine feedback instabilities; Section 5.3). However, large regions of the Antarctic coast that show large values of the CDF difference metric could not be exploited for model benchmarking simply because there are no rock outcrops in these regions.

*Criterion 2: Ice thickness CDFs diverge above the modern ice surface.* At least some of the differences between sensitized and desensitized model CDFs must occur at elevations *above* the modern ice surface so that corresponding ice cover frequency data can be collected without drilling through the modern ice sheet. For example, at the Whitmore Mountains in West Antarctica (Fig. 7f), ice thickness CDFs for the two models are different, but the differences are restricted to the lowermost elevations of the CDF, well below the present ice surface. A similar situation applies at the Ohio Range (Fig. 7b). Thus, these

sites are not useful because it would not be possible to collect data at the elevation range needed to differentiate models. This criterion is difficult to assess on a continent-wide basis, given (a) discrepancies between modern ice thickness and the ice thickness in the final timestep of our models, and (b) resolution issues when comparing average ice thickness within a 40km model grid cell with a sub-kilometre-scale nunatak. At some sites, bedrock samples near the present ice margin may be required to evaluate this criterion. For example, samples near the present ice margin at the Pirrit Hills show that the present ice thickness

is at the 20th percentile of the empirical ice thickness CDF (Fig. 9; see additional discussion in Section 6.3), which is much lower than would have been inferred from the "present" ice thicknesses in the model runs. Thus, coarse-resolution model simulations provide a guideline for applying this criterion, but additional information may be needed for some sites.



*Criterion 3: Ice thickness fluctuates significantly across glacial/interglacial cycles.* Differences between sensitized and
desensitized model CDFs must occur across a large enough elevation range to be detectable using ice cover frequency data
that could practically be collected. For example, the upstream Lambert Glacier location in Figure 7(c) has distinct ice thickness
CDF profiles, but these curves diverge across an elevation range of only 200m. Thus, collecting data that could differentiate
between these would require samples that were very closely spaced in elevation. Although most exposure-dating studies to
date have not collected closely-spaced data of this sort, it would likely be possible at some sites where bedrock in the needed
elevation range is extensive and accessible. However, it might not be possible at other sites if bedrock outcrops in the needed
elevation range were perpetually snow-covered, or too steep to access safely. Thus, model CDF predictions that diverge across
a large elevation range are more likely to be testable with data.

**6.2 Site selection criteria based on bedrock outcrop properties**

The criteria outlined in the previous section are derived from analysis of model simulations and describe locations where
geologic data could be used to distinguish sensitized and desensitized model simulations if suitable data existed at those
locations. However, additional geographic and geomorphic properties of bedrock outcrops dictate whether or not long-term
ice-cover histories could be reconstructed at these sites. In this section we consider field criteria for targeting sites for
model/data comparison.

*Criterion 4: Bedrock surfaces must record multimillion-year exposure histories.* In order to use cosmogenic-nuclide data to
reconstruct long-term average ice cover frequency, bedrock surfaces must preserve a long history of exposure. This requires
both low subaerial weathering rates during interglacial periods and negligible subglacial erosion rates during glaciations.
Existing exposure-age data from Antarctica show that in general, bedrock surfaces that record multimillion-year exposure
histories are common at relatively high-elevation nunataks in the interior of the ice sheet far from the coast (Fig. 8). On the
other hand, bedrock surfaces at lower-elevation coastal sites almost never record more than tens to hundreds of thousands of
years (Fig. 8). Thus, this criterion favours high-elevation, interior nunataks. High-elevation, interior sites are very likely to be
suitable for reconstructions of long-term ice thickness CDFs. Low-elevation coastal sites are not.

Bedrock surfaces with multimillion-year exposure ages have never been observed in coastal regions of West Antarctica, or in
the Antarctic Peninsula (Fig. 14). Despite the fact that end-member models predict highly divergent ice thickness CDFs
throughout much of West Antarctica, it is extremely unlikely that there exist any long-exposed bedrock surfaces in these
regions that could be used for model benchmarking as we propose here.

*Criterion 5: It must be possible to collect closely spaced samples across a large elevation range.* At a site where the model ice
thickness CDFs from sensitized and desensitized models diverge above the modern ice surface, it must be possible to collect





multiple bedrock samples within the elevation range in which they differ. This is easily achievable at ideal sites, such as the Pirrit Hills pictured in Fig. 4, where exposed bedrock extends from the present ice surface to well above the maximum height

ever covered by ice in the past, and bedrock at all elevations is ice-free and accessible on relatively gently sloping surfaces. On the other hand, it would not be achievable if, for example, model CDFs were different over a range of hundreds of meters, but exposed bedrock only extended tens of meters above the present ice surface. Even if bedrock did extend well above the present ice surface, it might consist of inaccessible cliffs, or small, widely separated outcrops separated by large elevation gaps. Insufficient relief or inaccessible bedrock would both make it impossible to collect data that could be used to distinguish

model results.

**6.3 Model benchmarking with ice-cover frequency data at the Pirrit Hills**

Here we discuss sites in Antarctica where cosmogenic-nuclide data exist that constrain the frequency distribution of ice thickness and therefore have the potential to distinguish between sensitized and desensitized models. The most comprehensive such data are from the Pirrit Hills, a nunatak group in the Weddell Sea Sector of West Antarctica (location in Fig. 7e). At that

site, Spector et al. (2020) measured $^{26}$Al, $^{10}$Be, and $^{21}$Ne concentrations in an elevation transect of bedrock surface samples collected between the present-day ice surface and the mountain summits 1000 m higher (Fig. 4). Figure 9 depicts these data, inverted for the fraction of time ice covered. Because these data collectively represent the portion of the ice thickness CDF above the present day ice level, they can be directly compared to model predictions of the same quantity (red and blue lines in Fig. 9).

At the Pirrit Hills, estimates of the percentage of time spent ice-covered decrease monotonically with elevation from ~80% near the modern ice level to values that are close to zero above a height of 400 m (Spector et al., 2020). The ice-thickness history implied by these data is supported by (i) glacial geologic observations, (ii) exposure dating of glacial deposits, and (iii) measurements on a subglacial bedrock core, which, together, establish that the ice sheet surface at the Pirrit Hills is nearly always between -150 and +400 m of its present-day level (Spector et al., 2019; Stone et al., 2019; Spector et al., 2020). The

total ice thickness variation implied by the sensitized and desensitized models is very similar to the observed range: both model CDFs show ice thickness ranging between -200 and +400 m relative to modelled present-day ice thickness.

The main challenge in establishing which modelled ice thickness CDF best fits the observations is determining what reference level to use for the present ice sheet surface. In Figure 9, we have referenced the model CDFs to the present ice thickness in each simulation. However, as noted by other studies that compare glacial geologic observations to ice sheet simulations (e.g.,

Briggs and Tarasov, 2013), the present ice thickness in a model is commonly not equal to the actual ice thickness. In part, this is because course-resolution models capable of million-year simulations cannot resolve small topographic features, such as the Pirrit Hills. Additionally, the models used here have not been specifically tuned to reproduce the present ice sheet geometry.



For these reasons, it is unclear whether the modelled ice thickness at present is functionally equivalent to the actual present ice thickness.

A workaround to this issue is to compare the shapes of ice thickness CDFs rather than their absolute values. This is done in Figure 10, which is identical to Figure 9 except the model ice thickness CDFs are offset in elevation such that observed and model CDFs are aligned at the 80th percentile of ice thickness – an arbitrary percentile but one that allows for visual comparison to the data. Figure 10 shows that, for all time periods, the shape of the desensitized model ice thickness CDF closely matches the empirical ice thickness CDF, while, in contrast, the sensitized model CDF has a distinct stepped profile

that is absent in the data. As discussed in Section 3, the differences between the two modelled ice thickness CDF shapes results from gradual versus rapid transitions between extreme ice sheet configurations in the desensitized and sensitized simulations, respectively. Thus, the empirical ice thickness CDFs from the Pirrit Hills are consistent with an ice sheet with weak marine ice margin instabilities. If replicated at multiple sites, this result would imply that the Antarctic ice sheet does not display very strongly nonlinear marine ice margin instability throughout the Plio-Pleistocene.

**6.4 Model benchmarking with other existing ice-cover frequency datasets**

The cosmogenic-nuclide measurements on bedrock surfaces from the Pirrit Hills are far and away the most comprehensive dataset from Antarctica that can be used for model-data comparison. This dataset also has characteristics needed for internal validation and assumptions testing, including measurements of three nuclides on samples from a closely-spaced elevation transect spanning the elevation range over which models predict ice-thickness variations. Other data sets from interior nunataks

have fewer data, sample a smaller range of elevations, or are discontinuously spaced in elevation. Some also lack model resolving power because they are located in areas where ice dynamics are not correctly resolved by the 40-km resolution model. For example, Balco et al. (2014) reported an elevation transect of multiple-cosmogenic-nuclide data from bedrock adjacent to Taylor Glacier in the Dry Valleys, that can be inverted for ice cover frequency. However, this glacier is not resolved in the 40-km model, so a model CDF for this site would be unrealistic. Regardless, we now review available data from other

possible sites. To identify other potential sites, we queried the ICE-D:ANTARCTICA database for locations having multiple-cosmogenic-nuclide data from bedrock samples, at least some of which yield apparent exposure ages of 1 Ma or older, and that span a range of elevations either on individual or closely spaced nunataks. We then applied the MATLAB code of Spector et al. (2020) to exclude samples demonstrably affected by erosion and, if possible, invert remaining data for ice cover frequency estimates. This yielded several candidate locations, as follows.

Spector et al. (2020) reported measurements of multiple-nuclide data on an elevation transect of bedrock samples from the Whitmore Mountains in central West Antarctica. These data demonstrate that the WAIS at this site has very rarely if ever been thicker than present (see discussion in Spector et al., 2020). Both sensitized and desensitized models are consistent with this result (Fig. 7f); thus the data are equally consistent with both models and the site has no resolving power.





A few paired $^{26}$Al/$^{10}$Be data from the Grove Mountains in East Antarctica (e.g., Fig. 7a) are inverted for ice cover frequency in Figure 11. However these data cover a very limited elevation range and are somewhat internally inconsistent, possibly due to the difficulty of relating data from several distinct nunataks, collected in different studies, to a common present ice margin elevation. More importantly, as discussed above, sensitized and desensitized model simulations yield very similar CDFs for this site, which would imply that even if more extensive data were available from this site, they would not be useful in

distinguishing the two models.

A few paired-nuclide data also exist for the Shackleton Range, which is a potentially valuable site because there are large differences between the CDFs predicted by sensitized and desensitized simulations (Fig. 7g,h). However, when inverted for ice cover frequency, these data are scattered and internally inconsistent. As at the Grove Mountains, this may be the result of geometric ambiguity in referencing data from multiple individual nunataks to a common representative present ice surface

elevation (also see discussion in Nichols et al., 2019). Alternatively, this site is coastal and at relatively low elevation, so bedrock erosion and weathering are likely. As the algorithm for identifying and discarding samples with significant erosion in Spector et al. (2020) is more effective for a denser elevation transect with more samples and ineffective when only one or two samples exist from the same nunatak, some of the apparent ice cover fractions may be biased due to unidentified episodic erosion. Regardless, it would be potentially valuable to collect a dense set of multiple-nuclide data from this region.

Mt. Hope sits at the mouth of Beardmore Glacier, which drains the EAIS though the Transantarctic Mountains into the Ross Sea, and is sufficiently large to be resolved by the 40-km model (Fig. 7i,j). Mt. Hope is promising for model-data comparison because (i) several bedrock samples from the upper flanks of the mountain have nuclide concentrations that indicate prolonged exposure and can be inverted for ice-cover fraction, and (ii) end-member model simulations predict CDFs with very different shapes. Unfortunately, as shown in Figure 13, the comparison is somewhat ambiguous with existing data. The empirical CDF

is more similar to the sensitized and desensitized model CDFs when integrated over the past 2.6 and 1.2 Myr, respectively, but given the small size of the dataset, neither fit is entirely compelling. Measurements on additional samples from this site could potentially help distinguish between model simulations.

To summarize, the existing data set that is best suited to comparison of model ice thickness CDFs with observationally derived long-term average thickness CDFs derived from cosmogenic-nuclide data is the set of multiple-nuclide data for the Pirrit Hills.

This data set is consistent with the desensitized model prediction, and inconsistent with the sensitized model prediction. However, although some similar data from other sites in Antarctica exist, they are either uninformative or ambiguous, primarily either because the data density or the elevation range of the data are inadequate, because sensitized and desensitized models do not make resolvably different predictions for the site, or, in many cases, both. Regardless, the potential significance of the



observation that the Pirrit Hills data strongly favour a desensitized model indicates that it would be valuable to collect
equivalent data from elsewhere in Antarctica. We now consider where this might be possible.

## 6.5 Where should we look next to infer past ice sheet behaviour?

The Pirrit Hills example in Section 6.3 shows that it is possible, in ideal circumstances, to collect geological data that provide
an empirical ice thickness CDF that can be used to differentiate between model predictions. In Section 6.4, we show that, at
present, there are no comparable data sets that are similarly useful. One reason for this is that many sites do not satisfy our five
criteria for sites where model/data comparison could be possible: ice thickness CDFs must diverge between sensitized and
desensitized ice sheet models at bedrock outcrop locations above the modern ice surface (Criteria 1 and 2); ice thickness must
fluctuate significantly across glacial/interglacial cycles (Criterion 3); and it must be possible to collect closely spaced samples
across a large elevation range where multimillion-year exposure histories are preserved (Criteria 4 and 5).

Another reason for a lack of comparable data sets has to do with the properties of the model and could potentially be addressed
with improved modelling efforts. There are many locations where cosmogenic-nuclide data now exist, or could be gathered in
future, in regions of complex topography where the 40-km model fails to resolve important aspects of ice flow. Many of these
sites are adjacent to glaciers in the Transantarctic Mountains that are, in reality, major conduits of ice from the East Antarctic
Ice Sheet into the Ross Sea, but are not large enough to be resolved in the 40-km model. These include data from Taylor
Glacier as mentioned above (Balco et al., 2014); Reedy Glacier (Todd et al., 2010), and Hatherton Glacier (Hillebrand et al.,
2021). These sites could be used for model-data comparison if the model resolution was increased sufficiently to correctly
resolve ice flow in these regions, perhaps by embedding a nested model domain in the low-resolution 5 Ma model runs.

The final reason is simply that data collection at many sites are very sparse. This can be addressed by additional field and/or
laboratory data collection. The Shackleton Range sites (Fig. 12) are an example of a location where some multiple-nuclide
measurements exist but the data are too sparse to use for model benchmarking. There are another 67 sites represented in the
ICE-D:ANTARCTICA database where multiple-nuclide data have been collected from bedrock, but only for one or two
samples at each site. Many of these sites do meet many or all of the criteria outlined above, so collecting denser and more
comprehensive data from these locations could potentially be valuable for model-data comparison. Here we briefly highlight
several of these locations.

*Shackleton Range* (Fig. 14, inset 1). Sensitized and desensitized end member models predict strongly contrasting ice thickness
CDFs above the present ice surface elevation (Fig. 7g,h, Fig. 12). Apparent exposure ages on bedrock exceeding 3 Ma are
known to be present (Fogwill et al., 2004; Sugden et al., 2014), so it is likely that many bedrock surfaces preserve long exposure
histories. Nunataks display several hundred meters of relief above the present ice surface. However, existing data comprise
only one or two measurements from each of several distinct nunataks. A disadvantage of this site is that the modern ice sheet
surface surrounding exposed nunataks is complex, as rock outcrops separate the high-elevation interior of the ice sheet from



much lower outlet glaciers, and, in addition, this area is remote. However, it appears possible that an effort to collect densely spaced elevation transects of bedrock samples from some nunataks in this region could yield an empirical ice thickness CDF valuable for model comparison.

*Lambert Glacier region* (Fig. 14, inset 2). End-member models predict distinct ice thickness CDFs above the present ice surface in this region (Fig. 7c,d), and, in general, the Lambert Glacier region shows a large divergence between models. Known

cosmogenic-nuclide data from bedrock samples in this region comprise only a few measurements, but some of them show multimillion-year apparent exposure ages (Hambrey et al., 2007; Lilly, 2008). This area is one of the closer areas of rock outcrop to the large subglacial basins in East Antarctica that are hypothesized to have deglaciated during past warm periods, and dense bedrock data from these sites may be useful for constraining models that do and do not predict such deglaciation.

*Wilkes Basin margin, northern Victoria Land* (Fig. 14, inset 3). Likewise, one of the key differences between end-member

model simulations is the extent of ice sheet collapse in the Wilkes Basin during warm interglacials, and this difference is clearly evident as the highest values of the difference metric D on Fig. 7 (and Fig. 6a). This would be one of the most valuable areas on the continent to be able to evaluate the model simulations. Although there are no ice-free areas within the centre of the basin where model differences are greatest, there do exist rock outcrops on the eastern edge of the basin, on the western edge of the northernmost Transantarctic Mountains. There are only three bedrock samples with cosmogenic-nuclide data in

this entire sector of the ice sheet (van der Wateren et al., 1999; Welten et al., 2008), but they indicate apparent exposure ages in the range 2-9 Ma, showing that low-erosion-rate bedrock surfaces are prevalent in this region. On the other hand, this is a region of complex ice flow, in which the presence or absence of ice in the Wilkes Basin is expected to force reversal of ice flow into or out of the Transantarctic Mountains, so it is likely that higher-resolution modelling would be needed to generate glaciologically realistic ice thickness CDFs. Thus, whether or not empirical ice thickness CDFs from bedrock elevation

transects in this region would be useful in constraining model marine ice sheet instability is more speculative, but the proximity of this region to the hypothesized location of significant ice volume loss in the Wilkes Basin means that data collection and high-resolution model simulations in this region would likely be valuable.

**7 Conclusions**

This work explores the use of long-term exposure age data from the Antarctic continent to differentiate between ice sheet

model simulations with stronger and weaker marine ice margin instabilities. We demonstrate that ice sheets with high parameterized sensitivity to marine ice feedbacks respond non-linearly to applied forcings and therefore spend more time in extreme minimum or maximum configurations, while desensitized ice sheets respond more linearly to forcings and spend more time in intermediate configurations (Section 3). These end-member simulations produce diverging characteristic patterns of ice sheet growth and decay (Section 4).
Ice thickness distribution over long timescales is both a model prediction and a geologic observation using cosmogenic nuclide concentrations at exposed nunataks. We compute and describe ice thickness cumulative frequency distribution (CDF) curves from both sensitized and desensitized model simulations (Section 5) that are directly comparable to long-term geologic reconstructions of ice cover frequency at any discrete location across the continent (e.g., Fig. 7). Ice cover frequency data from cosmogenic nuclide data can therefore be used to benchmark model simulations at suitable locations (Sections 6.1 and 6.2) to

infer past ice sheet sensitivity to marine ice margin instabilities.

We illustrate this model/data comparison approach at the Pirrit Hills, one of the very few existing transects of exposure age data across a sufficiently large elevation range along an interior Antarctic nunatak. The pattern of ice cover frequency at the Pirrit Hills is strikingly consistent with the desensitized model ice thickness prediction and inconsistent with the sensitized model prediction (Section 6.3). If replicated at multiple sites across the continent, this would be an extremely significant result,

implying that the Plio-Pleistocene geologic record provides evidence that the Antarctic ice sheet is not vulnerable to strongly nonlinear marine ice margin instabilities. However, other existing datasets from around Antarctica are either uninformative or ambiguous (Section 6.4). We therefore highlight targets for future geochronologic data collection (Section 6.5) to test whether ice sheet models that predict catastrophic sea-level impacts in future-analogue climates are accurate representations of past ice sheet change during warm periods.

**Data availability**

5 Ma sensitized and desensitized model simulations have been archived at the U.S. Antarctic Program Data Center (Balco et al., 2022a, b).

**Author contributions**

GB and PS conceptualized the project and provided funding. ARH and HB performed the model simulations. All authors

analysed the data and interpreted the results. ARH and GB prepared the manuscript, with contributions from all authors.

**Acknowledgements**

Funding for this project was provided by NSF (OPP-1744771 to GB and PS and OPP-PRF 2138556 to ARH) and by the Ann and Gordon Getty Foundation. We thank Dave Pollard for model development efforts that enabled these simulations, and for providing the data for Figure 1. Discussions with Trevor Hillenbrand improved this project.



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



(a)   (b)   (c)

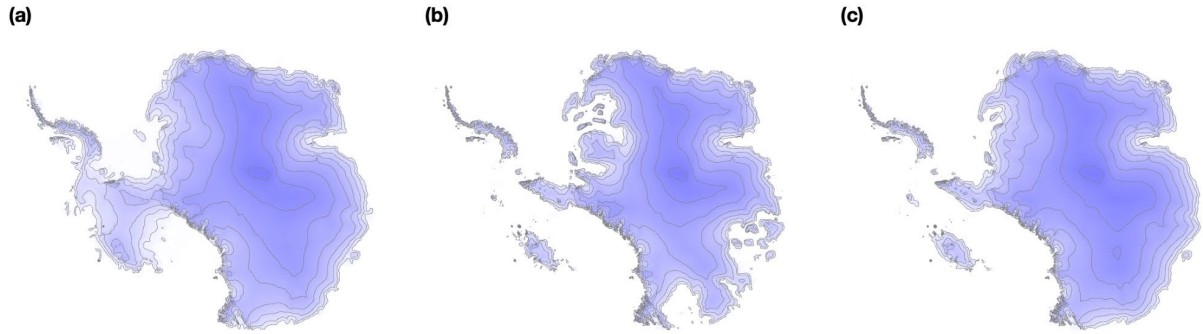

**Figure 1: Antarctic ice sheet model simulations with mid-Pliocene boundary conditions from Pollard et al. (2015), showing**
**distribution of grounded ice. (a) Modern ice sheet configuration used as starting condition for model runs. (b) MPWP simulation**
**with strong marine ice margin instability, showing extensive deglaciation of East Antarctic marginal basins. (c) MPWP simulation**
**without marine ice margin instabilities, showing minimal ice loss in East Antarctica.**



**Figure 2: Conceptual illustration of sensitized versus desensitized ice sheet behaviour. (a) For a given forcing (black bars), a desensitized ice sheet responds linearly (blue line) while a sensitized ice sheet responds nonlinearly (dashed red). The y-axis is shown as scaled deviation from the mean, so that ice volumes are normalized between -1 and 1 in this hypothetical illustration. In this example, the hypothetical dataset is non-linearized using a simple exponential transformation. (b) Frequency and (c) cumulative frequency distributions of ice volumes for the sensitized and desensitized models illustrated in (a). (d,e) Conceptual schematics of linear vs nonlinear responses to a given forcing (cf. Scheffer et al., 2009), leading to the differential ice sheet behaviour illustrated in (f,g). (f,g) Characteristic grounding-line behaviour in the Ross Sea, Antarctica; location shown as black box in (h). (f) Desensitized ice sheet behaviour is characterized by steady grounding line recession throughout a deglaciation, while (g) a sensitized ice sheet is more susceptible to runaway positive feedback mechanisms that cause the grounding line to rapidly jump from maximum to minimum states. (h) Continental bed topography (Fretwell et al., 2013) with a modern grounding line shown for context. (i,j) Characteristic ice thinning patterns for desensitized (i) and sensitized (j) ice sheet behaviour; coloured lines denote hypothetical ice surfaces as the ice sheet deflates and regrows over a nunatak, where cosmogenic nuclide data could be sampled along the modern exposed surface (k).**





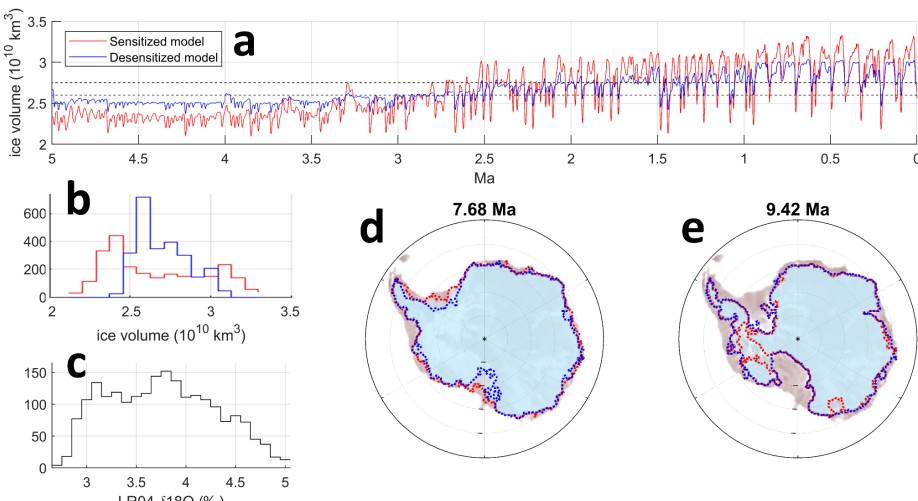

**Figure 3: (a) Grounded ice volume throughout the Plio-Pleistocene produced by the sensitized (red) and desensitized (blue) simulations. Dashed lines represent the modern ice volume and an approximate WAIS collapse threshold, respectively. (b) Histogram of ice volume fluctuations shown in (a). (c) Histogram of the d18O time series (Lisiecki and Raymo, 2005) used to force the model simulations. (d,e) Snapshots of sensitized (red) vs desensitized (blue) grounded ice configurations at two representative times; the sensitized ice sheet is bigger during glacials (d) and smaller during interglacials (e).**


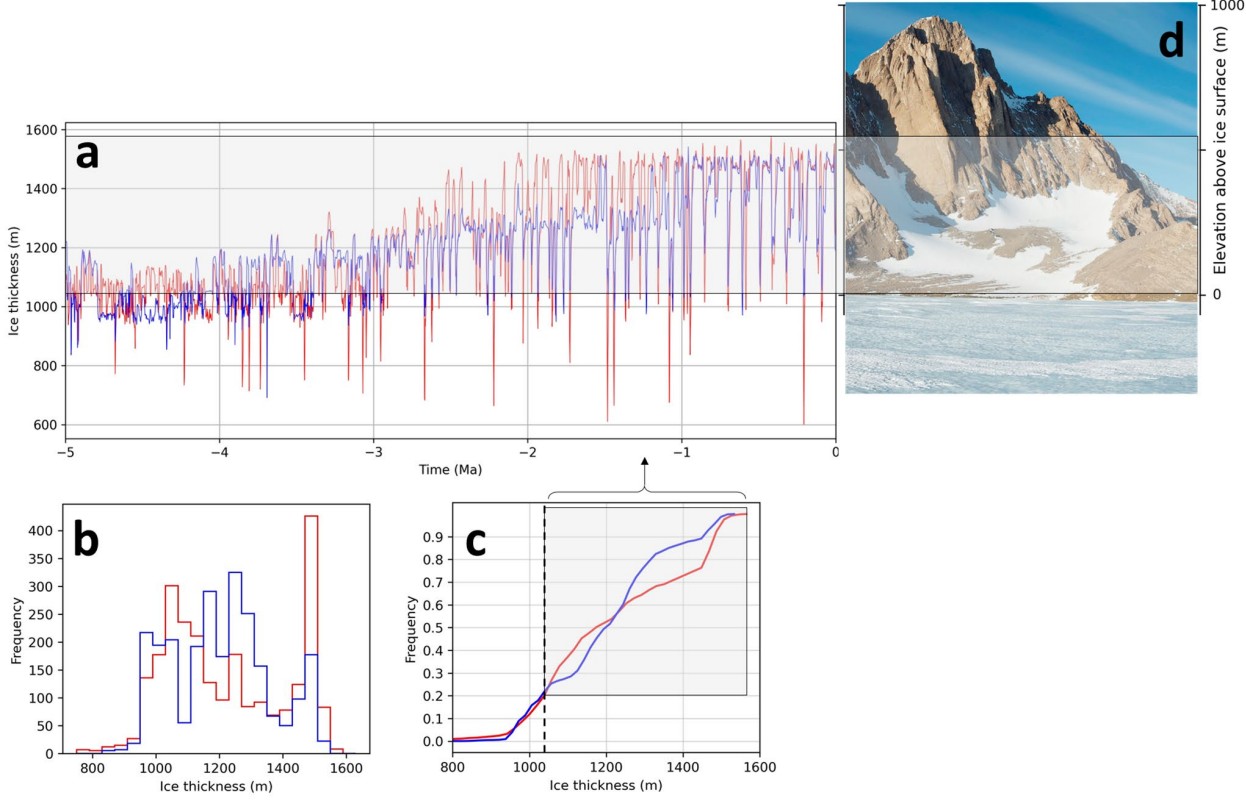

**Figure 4: Example ice thickness change history at Mt. Tidd, one of the nunataks comprising the Pirrit Hills, in the middle of the West Antarctic Ice Sheet (location shown in Fig. 7e). (a) The sensitized model (red) displays larger variation in ice thickness and is more likely to occupy extreme values, whereas the desensitized model (blue) is more likely to occupy intermediate values. The**
**resulting ice thickness histograms (b) and cumulative frequency distributions (CDFs) (c) are therefore distinct. (d) Photo of Mt. Tidd, aligned on the same yaxis as (a). Grey shading in (b,c,d) denotes the elevation range that is exposed above the present ice surface (so data can be collected) and where ice cover frequency reconstructions could distinguish between sensitized (red) and desensitized (blue) model behaviour. Dashed black line in (b) denotes the approximate modern ice thickness at this site. The dashed black line in (b) represents the modern ice thickness and is chosen to approximately align the range of ice thickness in the model**
**simulation with that inferred from geologic evidence.**



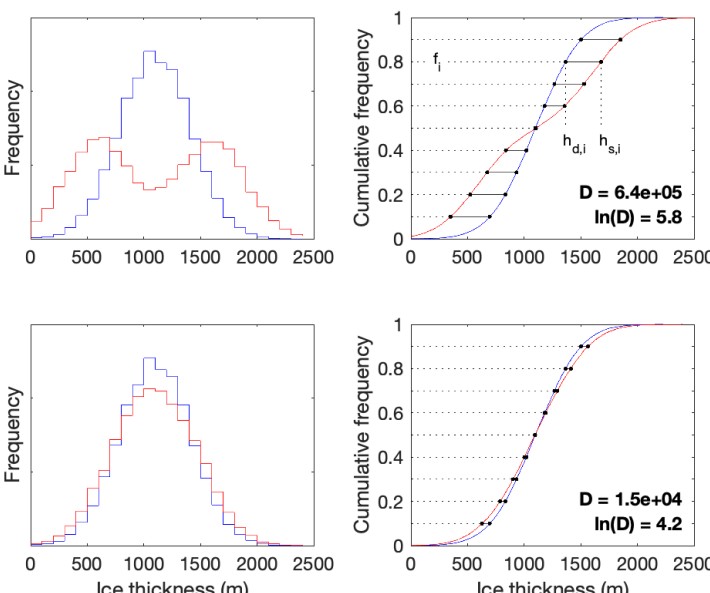

**Figure 5: Method of quantifying difference between model ice thickness CDFs at a site. Red and blue curves are hypothetical output from desensitized (blue) and sensitized (red) ice sheet model runs, displayed as histograms (left) and CDFs (right). The difference between two CDFs is quantified by sampling the difference between the elevations of the two CDFs at evenly spaced values of cumulative frequency (the black line segments in right panels), and computing a total CDF difference metric 'D' (see text) as the sum of squares of the individual differences. As D ranges over several orders of magnitude, for convenience we plot ln(D) in subsequent figures.**





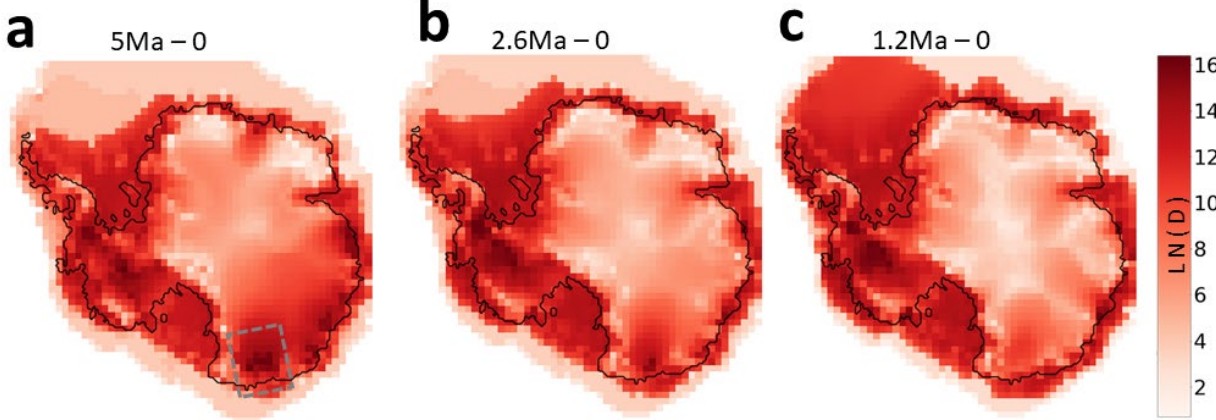


Figure 6: Spatial patterns in the CDF difference metric 'D' between our sensitized and desensitized model simulations (see Section 5.2). D is computed from model ice sheet evolution through the last 5 million years (a), the last 2.6 million years (b), or the last 1.2 million years (c). A modern model grounding line is shown in black. Dashed grey box in (a) denotes the Wilkes Subglacial Basin.







**Figure 7: Central plot shows the 5 Ma CDF difference metric ln(D) as shown in Fig. 6a, compared with areas where comparable geologic estimates of ice thickness CDFs could be collected. The green dots are locations where cosmogenic-nuclide data from bedrock at interior nunataks indicate exposure histories longer than 1 Ma, implying the possibility of generating observational ice thickness CDFs integrated over 1 Ma or longer. The surrounding plots (a-l) display ice thickness CDFs for the sensitized (red lines)** **and desensitized (blue lines) model at selected sites (azure blue dots) where some cosmogenic-nuclide data exist. The upper and lower Lambert Glacier, Beardmore Glacier, and Byrd Glacier (l) sites are representative rather than exact data locations, because the coarse resolution of the model means that existing exposure age datasets collected adjacent to these glaciers do not fall into the model grid cell corresponding to the glacier location. Thus, we plot ice thickness CDFs at a nearby representative grid cell. Dashed black lines are approximate (see discussion below in 6.2) representations of the present ice thickness at these locations derived from a** **reference model that reproduces the modern ice sheet configuration. Sites of exposure-age data represented by green dots are derived from the ICE-D: ANTARCTICA database.**



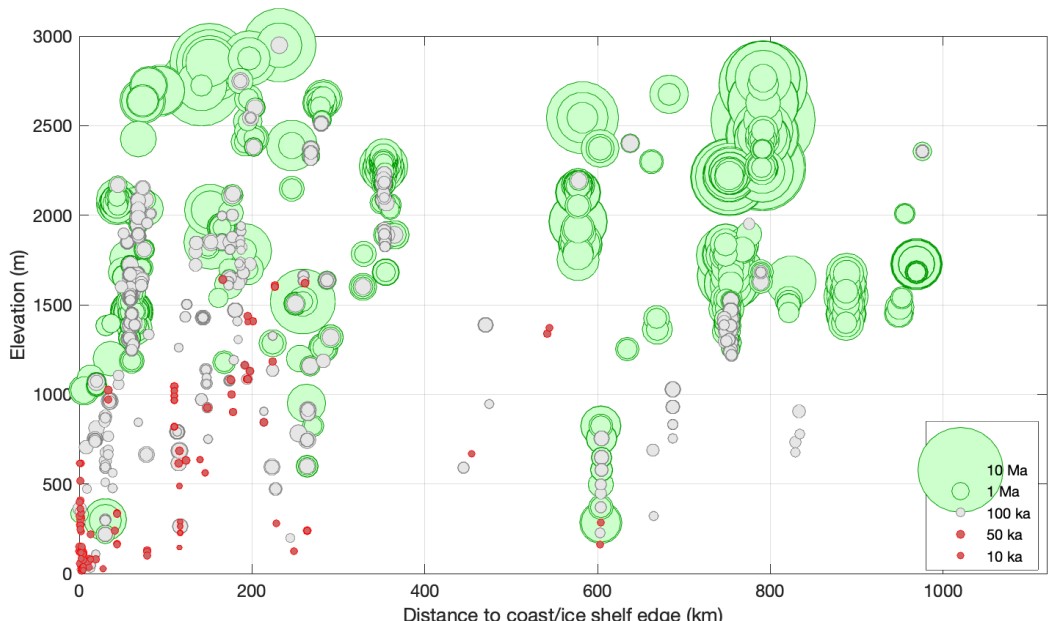

**Figure 8: Geographic distribution of bedrock exposure ages in Antarctica compiled in the ICE-D:ANTARCTICA database. Each circle represents a bedrock sample with at least one cosmogenic-nuclide measurement, and the size of the circle indicates the apparent exposure age of the surface calculated from that measurement. The "apparent exposure age" is the exposure age of the surface given the assumption that the surface has been exposed continuously for a single period. As the majority of these samples have been repeatedly covered by ice, the apparent exposure age is a minimum limit on the duration of the exposure history recorded by a sample. Green denotes samples with apparent exposure ages > 1 Ma, and red denotes samples with apparent exposure ages < 50 ka. Samples that record multimillion-year exposure histories are common at elevations above approximately 1500 m, are ubiquitous at high-elevation, inland locations, and are rare at lower-elevation, coastal sites.**





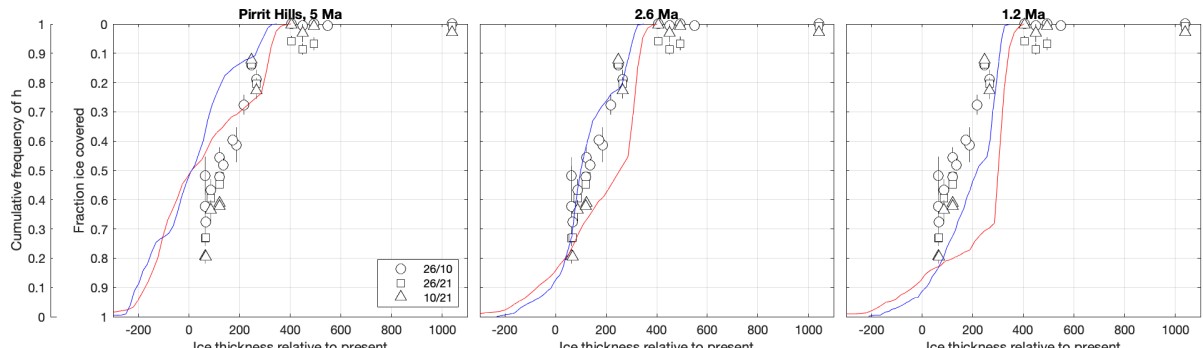

**Figure 9:** Ice cover frequency estimates derived from cosmogenic-nuclide data at the Pirrit Hills (Spector et al., 2020) compared with ice thickness CDFs for sensitized (red) and desensitized (blue) model simulations over different time periods. In this figure, model CDFs are vertically registered with data such that the "present ice thickness" is the ice thickness in the model grid cell containing the Pirrit Hills in the final time step of the model. The ice cover frequency estimates are computed from data in Spector et al. (2020) and the ICE-D: ANTARCTICA database, using the MATLAB code of Spector et al. (2020).





795

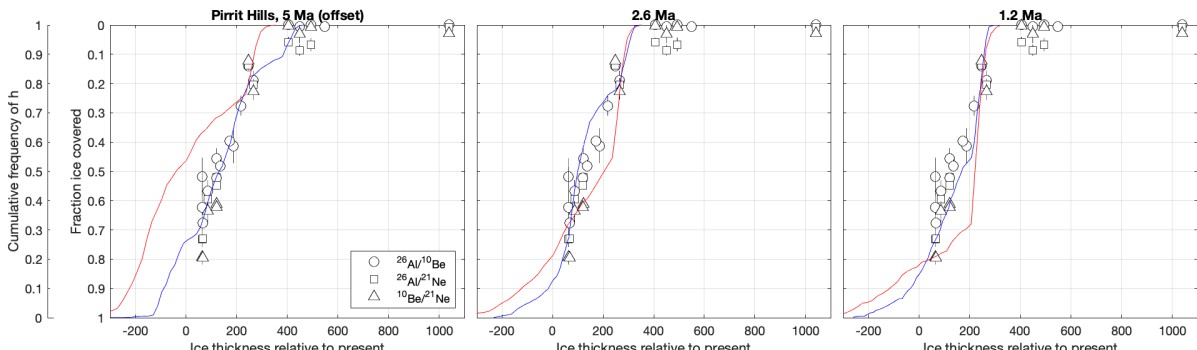

**Figure 10: The data in this figure are the same as in Figure 9, but the model ice thickness CDFs are offset in order to align observed and model CDFs at the 80th percentile of ice thickness.**



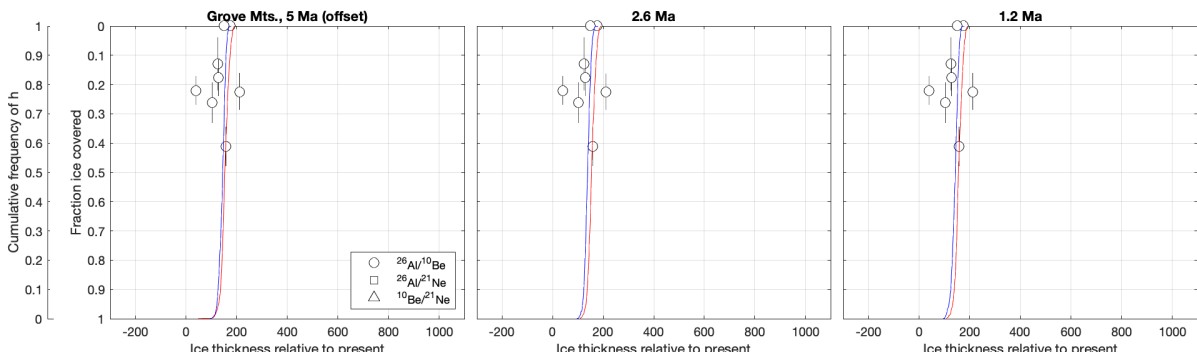

Figure 11: Ice cover frequency estimates for the Grove Mountains in East Antarctica inferred from paired $^{26}$Al/$^{10}$Be data and the inversion code of Spector et al. (2020), compared with sensitized and desensitized model CDFs for the same site. As in Fig. 10, model CDFs have been offset in elevation to align them with the centre of the group of data points. Data are described in Huang et al. (2008), Lilly (2008), Li et al. (2009), Kong et al. (2010), Liu et al. (201), Lilly et al. (2010). Sample elevations relative to the present ice margin are taken directly from the source publications without additional examination.





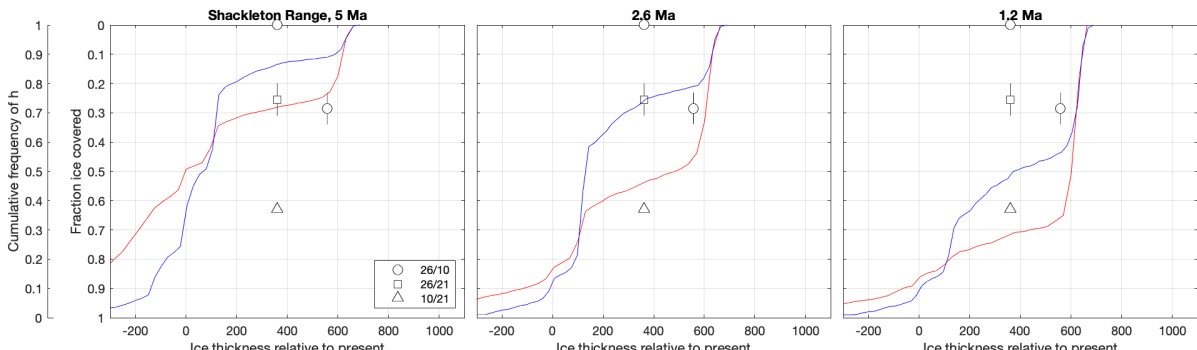

**Figure 12: Ice cover frequency estimates for the Shackleton Range inferred from paired $^{26}$Al/$^{10}$Be data and the inversion code of Spector et al. (2020), compared with sensitized and desensitized model CDFs for the same site. The model CDFs are referenced to the ice thickness in the final model time step and no model-data alignment has been attempted. The data are described in Fogwill et al. (2004), Hein et al. (2011, 2014), Sugden et al. (2014). Sample elevations relative to the present ice margin are taken directly from the source publications without additional examination.**

810





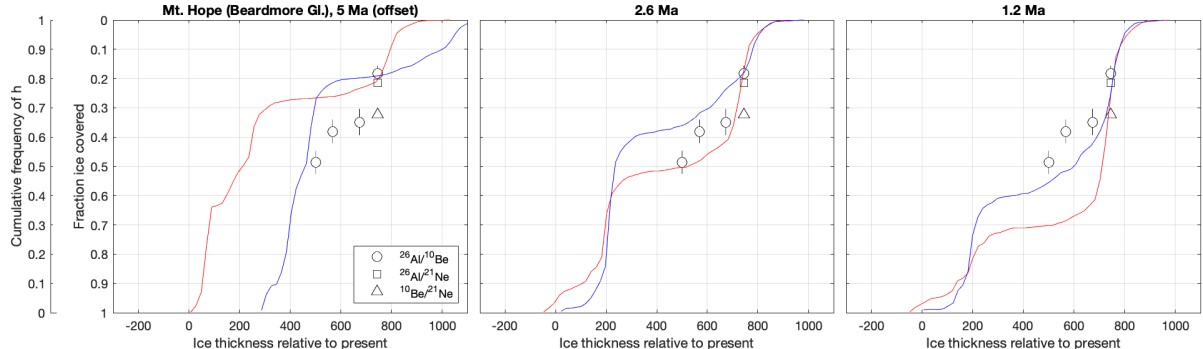

**Figure 13: Ice cover frequency estimates for Mt. Hope, inferred from multiple-nuclide data and the inversion code of Spector et al.**
**(2020), compared with sensitized and desensitized model CDFs for a site in the centre of the model Beardmore Glacier at a position**
**near Mt. Hope (because of the coarse model resolution, the model Beardmore Glacier is not in exactly the same location as the real**
**glacier, so we have chosen an equivalent site in the model). As in Figs. 10 and 11, model CDFs have been offset to align observational**
**and model CDFs at the 80th percentile elevation. The data from Mt. Hope are unpublished measurements archived in the ICE-**
**D:ANTARCTICA database.**



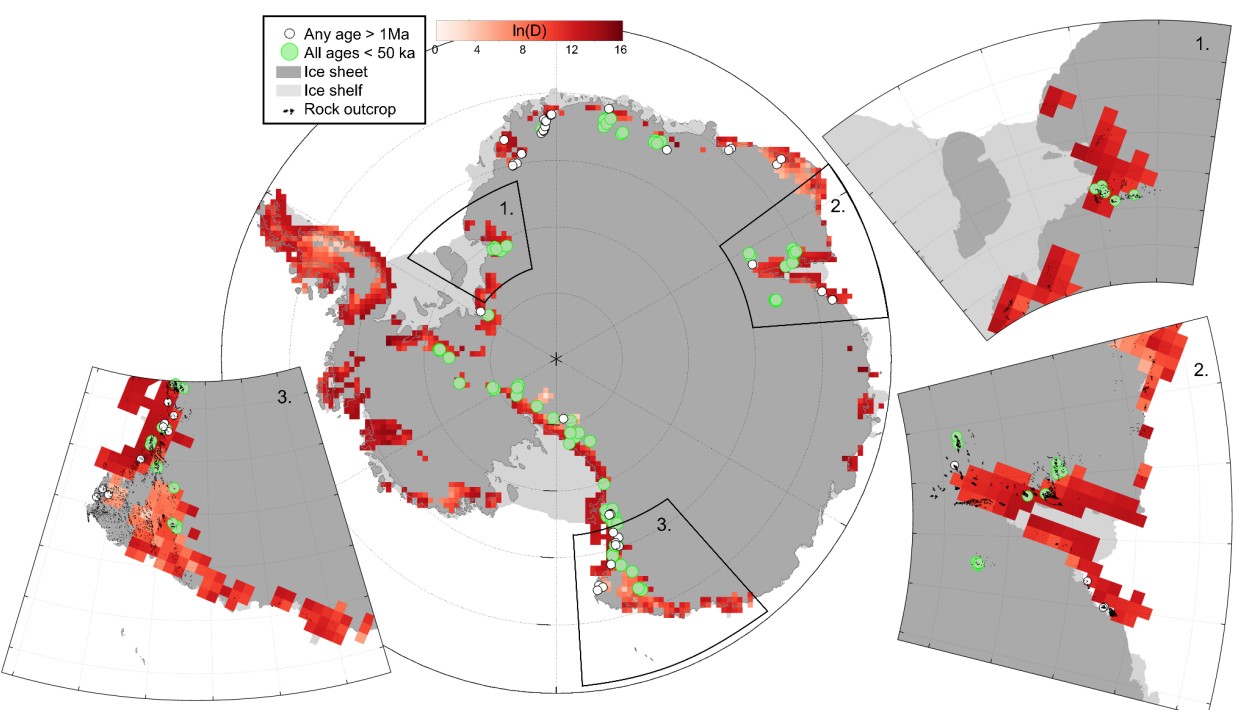

**Figure 14: CDF difference metric D at possible locations for model/data comparison; D is cropped to locations within 50km of rock outcrops, and where total modelled ice thickness changes are greater than 400m (following Criterion 3). Existing exposure age samples are plotted according to age so as to highlight sites where long-exposed bedrock surfaces are (green dots) and are not (white dots) likely to exist. In general, ice-free areas where apparent exposure ages > 1 Ma have been observed (green dots) are located in relatively high-elevation, inland regions (e.g., Criterion 4). Sites where bedrock exposure-age data have been collected, but only relatively young exposure ages have been observed, are in low-elevation coastal regions where subglacial and subaerial erosion are more likely to occur. Modern ice sheet and ice shelf extent from Fretwell et al. (2013). Inset panels 1,2,3 show Shackleton Range, Lambert Glacier region, and Wilkes Basin margin/Northern Victoria Land, respectively, with rock outcrop locations in black.**