# Peer review of "Cosmogenic-nuclide data from Antarctic nunataks can constrain past ice sheet instabilities"

_The Cryosphere, 2022_

## Referee Comment (RC2)

Comment on "Cosmogenic-nuclide data from Antarctic nunataks can constrain past ice sheet sensitivity to marine ice margin instabilities" by Anna Ruth Weston Halberstadt et al., (https://doi.org/10.5194/tc-2022-213)

The paper investigates the use of cosmogenic nuclide measurements from nunataks across the interior of Antarctica to evaluate the performance of ice sheet model simulations over past time periods. Under certain assumptions and requirements, the inferred frequency at which the sample sites have been covered by ice can be used as a proxy to determine a cumulative distribution function for ice sheet thickness in the surroundings of the nunatak. Uncertainties aside, the same quantity can be derived from an ice sheet simulation for that location, and thus a direct comparison between model and data is possible. In addition, the authors propose a metric based on the model version of this function to identify regions where different model realisations exhibit large discrepancies in their integrated behaviour. According to the authors, future measurements over some of these regions could provide valuable benchmarking capabilities for ice sheet models used in both paleo reconstructions and future projections.

In my opinion, model-data integration is a topic in ice sheet modelling that has been undeservedly left as a secondary problem due to computational limitations that, so far, have made it difficult to create bridges between the field and modelling communities. As shown by the authors, the method explored in this study is a valuable approach that can, in principle, be used to evaluate virtually any modern paleo ice-sheet set-up. As more precise and abundant field data are gathered, and as the capabilities of state-of-the-art ice-sheet models increase, the applicability of this method will likely improve. As such, the core of the study is a nice addition to the scientific literature and, in this sense, I support its publication in TC. However, there are several points which I found problematic in the way the authors chose to showcase the applicability and significance of the method, particularly within the modelling set-up. I think that these choices do affect the conclusions, and that confirmation that this is not the case is needed before acceptance of the manuscript. I will describe these points as part of my general comments below, and if necessary elaborate on them in the specific comments after that.

General comments:

1. As the authors acknowledge ("Criterion 1" in Discussion), the model simulations need to have sufficiently different CDFs for the method to be usable as a constraining tool. The authors then generate these two sufficiently different model realisations by including/excluding a representation of the ice shelf hydrofracturing (HF) and marine ice cliff instability (MICI) processes, and by simultaneously scaling up/down the time-dependent ocean temperature shift that the model uses to derive melt rates at the base of ice shelves. To clarify my concern, and to prevent specifics worries being diluted in a single comment/response, I will divide it into several smaller points:

A. The authors put both HF and MICI in the same category as the stronger/weaker ocean forcing, and name them all "marine ice margin instabilities". I agree with HF and MICI, as these processes are inherent to ice dynamics, i.e., the modelled ice is reacting (or not; through crevasses and calving) in some consistent way to some time-varying external

boundary conditions (e.g. surface temperatures, meltwater ponds, accumulation). If a fixed change in the underline{external} forcing is applied, both model realisations will react very differently, with the one including both HF and MICI potentially generating positive feedbacks that can trigger significant ice sheet retreat (e.g. [1]). In other words, a model with HF and MICI *is* more sensitive to a given change in the external forcing.

B. However, applying different ocean temperature shifts is no more than simply applying different changes in the external forcing, not inherent to ice dynamics: apply the same change in ocean temperatures (the underline{external} condition) to both models, and the resulting thinning/thickening of ice will be the same. Here, ocean temperature reflects an uncertainty not in model sensitivity, but in the external forcing. In other words, both models are *equally* sensitive to a given change in ocean temperature, and thus equally (un)stable and their responses equally (non-)linear. In this regard, the study is not constraining marine ice instabilities, but simply stronger vs. weaker forcing conditions.

C. A way to include different sensitivities to a given ocean temperature change would be to use two distinct ocean melt parameterisations, e.g. one where total melt is linearly proportional to ocean thermal forcing and another that is quadratic. Those two will react very differently to, say, a 1 K change in local ocean temperatures. Another would be to test melt under partially floating grounding line cells vs not doing it (e.g. [2]), which as a bonus is, in my opinion, equally controversial as HF and MICI. Both options work directly on an inherent ice dynamics quantity, namely dH/dt, instead of the external forcing. That I would consider closer to a "marine ice margin instability", together with HF and MICI.

D. Why such a strong difference in the two end-member simulations? HF and MICI alone (albeit only in tandem) were enough to generate a huge difference for the Pliocene in [1]. Adding on top of that two full degrees Celsius to each side of the ocean temperature shift for the sensitised model feels overkill. And that's not even considering the extra warming added during interglacials to the Amundsen Sea area. I imagine that the method would struggle with less extreme differences, i.e. with fulfilling Criterion 1. What happens when only HF and MICI are used, but not different ocean forcings? Are the resulting CDF's still different enough to make the method usable? Could it be that the sensitised CDF fit is poorer than the desensitised one because of this extreme ocean influence? See also my specific comment L38-39 below.

E. I have the inkling that the ocean forcing is dominating the apparent bimodality of the sensitised CDF, which for me would be problematic, since i) I consider it as external forcing, not as an instability, and ii) it is, as far as I can tell from the manuscript, heavily parameterised. If this is the case, then the method is not constraining marine ice margin instabilities, but simply which parameterised forcing is less unrealistic.

F. A way to confirm this would be to run a third simulation with HF/MICI, but with the same ocean as the desensitised run (i.e. [-1 1]), and a fourth simulation with the ocean full temperature difference applied (i.e. [-3 3]), but with no HF/MICI. That would provide a clearer way of interpreting the model results, and –more importantly-- an assessment of the applicability of the method.

G. If my concerns are confirmed, most of these issues can be alleviated a bit by toning down the title and avoiding specific claims, e.g., "ice sheet sensitivity to marine ice margin instabilities" -> "strong vs weak ice-sheet retreat and advance histories".

2. While reading the paper, it is not fully clear to me what exact version of the PSU3D ice sheet model the authors used, and therefore how different processes are exactly represented in the study. The authors first introduce (L28) the model with some results from the 2015 version [1], but then cite (L169) the 2021 version [3]. Then, they mention (L172) that the approach in the 2009 version [4] is needed for the external forcing, for which I assume they use a modification of it since modern data sets of surface air temperature, precipitation, and subsurface ocean temperature are here mentioned (none of which is part of [4]), but this modified approach is only modestly described, and without any formulae (e.g. L173 "scaled based on a combination of factors"). Then, an additional parameterisation is mentioned for ocean temperature shifts (L175). Finally, L191 mentions an additional modification for the Amundsen Sea, only for the sensitised model, and only during interglacials:

A. In this version of the manuscript, it is not possible to get a clear idea of the forcing implemented. An appendix is needed where this and every other change with respect to [3] is provided clear and transparently. These details are important since the conclusions might depend on them (see my general comment 1).

B. With the above modifications, this is a different model and thus it does not make much sense to reference previous studies in terms of specific model behaviour. The introduction mentions the difference between models with and without HF and MICI (L30 onwards), which sets the ground for the rest of the paper in terms of instabilities, retreats, sea level contributions, and the inability of previous attempts at discerning between such models (L35). Finally, the goal of the paper and the sensitised vs. desensitised models are introduced (L38-42) within this context. This whole introduction heavily uses Fig 1, which represents data not from this study, but from [1]. After the significant work that running 5 Myr simulations I assume implies, which contain almost all periods covered by [1], [3], and [4], and the fact that this is a different model version, I expect the authors to illustrate their points, and in particular Fig 1, using the output from the model version used in this study. This will be a great chance to showcase the performance of both simulations for not only the Pliocene and present-day, but other crucial benchmark periods, such as 125 ka and 21 ka. See my specific comment L170-171 below.

3. Missing critical reference for the insensitivity of results to model resolution (L171-172). I assume you want to cite one of the model intercomparison initiatives where results seem to suggest that coarse grid resolution models can bypass the 100s-of-meters grounding line resolution requirements, when using e.g. Schoof's flux parameterisation or a sub-grid interpolation of basal friction at the grounding line (I am also assuming you are using the former). However, those intercomparison exercises usually employ idealised geometries that are far from the steep topographic features found in Antarctica, especially around nunataks:

A. I'd like to see some kind of indication that the results presented in this study and/or the version of the model used by the authors are/is resolution-independent under a similar Antarctic set-up.

B. If such an indication does not currently exist: I understand that 5 Myr is in general computationally expensive, but comparison against sub-set periods, say, even 500 kyr at 32/20 km and 250 kyr at 16km (assuming a ~ten-fold increase in comp. time per doubling of resolution) should allow you to confirm the claim in question, at least regarding the scope, assumptions, and conclusion of this study. This is valid even when the geological data would require longer time spans: the goal is to check how different are the CDF's for limited sub-set periods between simulations at varying resolutions, and not against the data.

Specific comments:

L26 – Could you please elaborate on what you mean by "ambiguous", and add it to the text? The word is used a few times throughout the manuscript, but the only time it is elaborated is in L451, where data are not dense enough (that would be insufficient) or taken from an approach-irrelevant elevation (that'd be uninformative). That is not a problem of the data, though.

L28 – Maybe a matter of opinion, but I would argue that is the present climate (~415 ppmv, ~1 ºC over Pre-Industrial) state the one approaching the peak mPWP states (~350 - ~450 ppmv, ~2.0 - 4.0 ºC above Pre-Industrial, inc. higher sea level), at least when acknowledging the poor paleorecord, uncertainties, and biases. Perhaps removing this sense of direction altogether? References [5] and [6] below (and therein) contain classic, neutral examples of this comparison.

L30 – This would be a good place to introduce the concepts of ice shelf HF and MICI, after "… ice margins (Fig.1)" and before "Model runs…". These concepts are used extensively in this study, and it would be nice to minimally describe them here. Otherwise, the reader is left hanging until L43, where "ocean temperature" is added to the mix (see my general comment 1), only HF is mentioned (not described), and where my brain predicted MICI to be also mentioned (but it was not). This is because [1] show that, at least in their highly parameterised and somewhat speculative (their words) simulations both HF and MICI are needed in tandem to get a retreat similar to that shown in Fig 1c. The reader is then left hanging again until some more info is given in section 3.1, which was in my experience distracting.

L38-39, L56 – This connects to my general comment 1. If the aim of the paper is to explore how to differentiate between margin instabilities, and Fig 1b and 1c are given as examples, then by using buffed and nerfed ocean shifts compared to the homogeneous +2 K that [1] used for their Pliocene, then the answer to the interesting question is obscured: "If we adapted both simulations F1b and F1c in [1] to a 5 Myr run (i.e. turning [1] into [4]), would we be able to discern between them using geological data and CDF's?" That's a beautiful question, but by modifying the ocean the answer is biased.

L71 – Section 2 was a pleasure to read.

L129 – There it is: "given the same external temperature and accumulation forcing". I fully agree.

L135 – "By a more linear", perhaps? There and elsewhere

L136 – endmember -> end-member (e.g. L124)

L138 – "Gray bars", maybe?

L148, L161 – Here MICI is mentioned, but not ice shelf HF. See my comments above on "Line 31". Consistency is needed on how you define and utilise these terms throughout the manuscript.

L150 – See my general comment 1, and specific comment L39-39, L56 above. Marine ice sheet instability (MISI) is not something that you can (or rather should) "turn on or off" in ice sheet simulations; it is a consequence of a retreating grounding line on a retrograde bedrock slope. If you try to supress it by using colder ocean temperatures during warm periods (colder than the +2 K needed and justified by [1] themselves), and then concluding that geological data support this, that's a very bold statement.

L162-165 – That's a long parenthesis. I'd suggest let it be its own full sentence.

L170-171 – See my general comment 3.

L172-180 – See my general comment 2A. It would be much clearer to add an appendix with the exact formulations (i.e. in equation form) mentioned/used in this study.

L173 – Which modern forcing data sets? Model-based (e.g. RACMO, MAR) or observational (e.g. WOA)? See my general comment 2.

L181 – Do you limit the lower shift in ocean temperatures based on the local pressure-freezing point of sea-water? Otherwise (assuming no supercooled water processes are resolved by the model) you could be assuming a below-freezing ocean in many areas during glacials. That would create mass "out of thin water", artificially making your ice sheet advance and your ice shelves proliferate.

L184 -- Same as specific comment L162.

L186 – Missing parenthesis at the end of the TraCE reference

L187 – I guess you mean "desensitized" here.

L189-190 – This is key to avoid model biases contaminating your conclusions, and it surely needs a figure supporting this claim. The easiest would be to show Pliocene (classic), 125 ka (recent interglacial example), 21 ka (recent glacial example), and transiently reached 0ka,

for both simulations. A figure like that would be much more informative than current Fig. 1. See my general comment 2B.

L191-196 – See my general comment 1. As it is surely evident by now, I am not easy with forcing the models by using different ocean temperature shifts and calling that a margin instability. Here, an even stronger difference is applied between the two simulations. What happens without this 1.5 K extra increase? How is the factor applied only during interglacials: as a shock once a time-point is reached, or smoothly as the interglacial is approached? Why is that trend not already in the modern ocean data set used as forcing? Even within quotes, a +1.5 K warming is hardly noise for a simulation that is limited to a modest +1 K above modern.

Lines 197-208 – Again, a summary of the mathematical expressions of the code for key processes discussed in the paper (e.g. atmosphere/ocean forcing and how it evolves, ice shelf HF, and MICI, …; i.e. all about how both simulations differ) is a currently missing appendix. Otherwise, the reader is forced to navigate through the last 10 years of PSU3D development without a clear idea of the precise version of the model used in this study.

L220-222 -- See my general comment 1. I would argue that using different ocean temperatures in your simulations qualifies as "a property of the forcing dataset", which could be in fact playing a primary control on your characteristic model behaviour. By the way, if I understand your method, d18O is not the only forcing used, but insolation and CO2 are involved as well. It'd be nice to have frequency distributions for those as well, and a distribution of the final weight used in the forcing parameterisation. I suspect the additional scaling added to the ocean temperature shifts, which differs between the models, could be bimodal as well. In other words, a frequency distribution of the desensitised ocean temperature shift would be more centred around 0 (by definition, since it is capped between -1 and 1 K) than the sensitised one, which would sit more often in either of the extremes. In that case, I would think that the bimodality of the sensitised simulation is strongly correlated to the ocean temperature applied, which in this case I consider part of the forcing, rather than an ice dynamical instability represented in the model. Could you please confirm this?

L223-228 – I am not convinced. See my general comment 1. Even if the simulations resemble the conceptual example, this could be a "self-fulfilling prophecy" case, with the external ocean forcing the main suspect.

L242 – "desensitized" perhaps?

L261 – MICI not mentioned.

L268 – You mean Fig. 6c, maybe?

L294 – "Previously" not needed, in my opinion. Sounds to me like a past publication.

L377 – "inverted for the fraction of time ice covered". Feels like something is missing here.

L378 – What time period do these data cover? Are you comparing it to the model CDFs computed for the same period?

L392 – That's confusing. Then against what were the models calibrated?

L400 – "differences", thus "result"

L403-404 – Bold statement. Difficult to assess without a clear quantification of the isolated impacts of a highly uncertain ocean forcing. Even if that were the case, then the desensitised simulation would need to be further evaluated against several other constraints (e.g. [7]), since it was "not benchmarked against any geological or glaciological data" [8].

L407, L409 – Consistency using "data set" vs "dataset". Here and elsewhere.

L417 – "Matlab code" sounds too specific, what about "approach" or "algorithm"?

L421 – "WAIS at this site" -> "this site", maybe?

L421-422 – I would say "and has, in this case, no resolving power". There is always the chance that in the future, a model will be able to better resolve this area, and perhaps generate distinct CDF's. Same with L429.

L445 – If I understand the method correctly, you should choose an integration time matching the field data temporal span, or?

L453 – Although (to my surprise) it exists, "resolvably" does not seem to fit the context here.

L472 – "Collection" is singular, thus "is", perhaps?

L509/510 – Not sure that the data are used to differentiate between the two models, as the models can be separated without the data. Maybe "evaluate the performance"?

L522-526 – I think that the strength of these claims needs to be checked against my concerns above. In any case, and before such a hypothetical strong implication (L524) could be trusted, a significant amount of validation would be needed. Considerably higher resolution models, with the ability to zoom in around outcrop areas and correctly approximate the dynamics of ice in high complexity topographies, would be first. A way more realistic external forcing would be needed as well. And then, validation of the results against a myriad of other geological constraints and modern observations would be required, especially if the validation of the model has the ultimate goal of enabling a simulator of future ice sheet dynamics.

L546, L548: links seem unnecessarily complex: https://doi.org/https://doi.org/

Figure 1 – This figure can be much, much improved. As a minimum, I would expect a way to quantitatively interpret the contours and colormap, including the name of the plotted quantity (ice surface elevation I assume). Maybe a consequence of the colormap choice, but where are the ice shelves extents and their thickness in subplot a (and presumably b and c)? Modelled boundaries (grounding line and calving front) would help too. Adding the present-day observed grounding line and calving fronts in a distinctive manner, to allow for a quick evaluation of the model's ability to reproduce the present state of the ice sheet, would be useful to the reader as well. You can optionally take the opportunity to add some general Antarctic locations here as referenced in the paper (for the reader's convenience), as precise locations are introduced only starting on Fig 7, central panel. By the way, the caption of Fig 4 mentions that the location of Pirrit Hills is shown in "Fig. 7e" (instead of central panel), which is confusing as it seems that you are using the lowercase letters to reference both locations on a map and subplots (with the latter being wrongly referenced anyway). In any case, I'd rather have it replaced by a figure showing fata from the simulations carried out in study (see my general comment 2B and specific comment L189-190).

Figure 2 – Those bars are grey, in my opinion. That rectangle with the arrow is black, but I am not sure about the need for it, or its accuracy: the subs d and e do not seem to correspond. Figure needs more love: It seems to me that you forgot to correctly white-rectangle-cover the "5"s in the horizontal axis of subfigure b. "Scaled" is not capitalised in subfigures a and b; same with "time" and other words. Please make the capitalisation consistent across all figures. Style of "(f,g)" not consistent, recommend using "Desensitized … (f), while sensitized … (g)". Subfigures f and g are counterintuitive: the linear response shows actually more retreat than the nonlinear one, which contradicts the rest of the paper. What process would cause something like that? Is the figure made up or from the simulations? Subs i and j would benefit from using the same colour scheme, preferably colour-blind friendly, to show corresponding surface levels.

Figure 3. What are the dates in d and e? I thought the model runs between 5 Ma and present. Are these from the current study? What is the y-axis in b and c? Some figure details need a minimal explanation in the captions, otherwise they assume the reader knows already those details.

Figure 4 – Photo credits? You might get some copyright warnings later.

Figure 6 – Why is D so big over the continental shelf and beyond? Does the ice reach the model domain boundaries?

Figure 7 – Why the (l) only for Byrd in the caption?

References:

[1] David Pollard, Robert M. DeConto, Richard B. Alley: Potential Antarctic Ice Sheet retreat driven by hydrofracturing and ice cliff failure, Earth and Planetary Science Letters, 2015, https://doi.org/10.1016/j.epsl.2014.12.035.

[2] Seroussi, H. and Morlighem, M.: Representation of basal melting at the grounding line in ice flow models, The Cryosphere, 12, 3085–3096, https://doi.org/10.5194/tc-12-3085-2018, 2018.

[3] DeConto, R. M. and Pollard, D.: Contribution of Antarctica to past and future sea-level rise, Nature, 531, 591–597, https://doi.org/10.1038/nature17145, 2016.

[4] Pollard, D. and DeConto, R. M.: Modelling West Antarctic ice sheet growth and collapse through the past five million years., Nature, 458, 329–332, https://doi.org/10.1038/nature07809, 2009

[5] Haywood, A., Dowsett, H. & Dolan, A. Integrating geological archives and climate models for the mid-Pliocene warm period. *Nat Commun* **7**, 10646 (2016). https://doi.org/10.1038/ncomms10646

[6] Stokes, C.R., Abram, N.J., Bentley, M.J. *et al.* Response of the East Antarctic Ice Sheet to past and future climate change. *Nature* **608**, 275–286 (2022). https://doi.org/10.1038/s41586-022-04946-0

[7] Ely, J.C., Clark, C.D., Hindmarsh, R.C.A., Hughes, A.L.C., Greenwood, S.L., Bradley, S.L., Gasson, E., Gregoire, L., Gandy, N., Stokes, C.R. and Small, D. (2021), Recent progress on combining geomorphological and geochronological data with ice sheet modelling, demonstrated using the last British–Irish Ice Sheet. J. Quaternary Sci., 36: 946-960. https://doi.org/10.1002/jqs.3098

[8] Balco, G., Buchband, H., and Halberstadt, A. R. W.: 5 million year transient Antarctic ice sheet model run with "desensitized" marine ice margin instabilities, https://doi.org/https://doi.org/10.15784/601601, 2022a.
* * *
Jorjo Bernales, 01.01.2023.

---

## Author Comment (AC1)

**Response to review 1 of 'Cosmogenic-nuclide data from Antarctic nunataks can constrain past ice sheet sensitivity to marine ice margin instabilities' by Halberstadt et al.**

This is a well-written and interesting manuscript comparing long-term ice cover histories derived from multiple cosmogenic nuclides measured in slowly eroding Antarctic bedrock surfaces with two ice-sheet model parameterizations. The manuscript suggests a novel way of testing these ice-sheet model parameterizations by use of empirical data. The paper is of wide interest and convincingly written. My comments below mainly contain suggestions to improve/clarify the figures and figure captions.

*We thank the reviewer for the supportive comments and helpful suggestions. Below we respond to reviewer suggestions and concerns in detail; for typographical errors or small corrections, the * symbol indicates our intention to enact these corrections in a revised manuscript. We appreciate the reviewer's attention to both the large and small aspects of our manuscript.*

Figures

Fig. 2:

- The box in panel (a) is not explained in caption, I assume it is outlining the extend of boxes in (d) and (e), but if so, the blue curve in (d) is lacking an initial and partially a final 'plateau' (horizontal part) similar to the red curve in (e). Even though it is a conceptual figure, I think this way of exaggerating the difference is misleading.

*Thanks for pointing this out. Boxes (d,e) were intended to be conceptual, but we agree that the box in (a) is not representative of these blue and red curves in (b,c) and doesn't make a lot of sense in this context - we will remove the box, leaving the rest of (b,c) as is.*

- Are you sure the blue curve in panel (b) is reflecting the blue curve in panel (a) correctly? Comparing the curves in panel (a), the blue and red curve shapes are similar – the blue curve has lower amplitude, but still quite steep slope around zero crossings. I would therefore expect the blue curve in (b) to be bimodal, but with less sharp peaks located closer to zero compared to the red line (approximately +- 0.25).

*The frequency distributions in (b,c) are indeed calculated directly from the red and blue curves in (a). "Steep" is relative, and the red curves are many times steeper than the blue curves at the zero crossings, which gives rise to the difference in the frequency distributions. For example, zooming in to just a few "glacial cycles" (shown below in yellow) reveals that the blue curve indeed does spend more time closer to zero.*

[Figure]

- Scalebar for panel (b) is missing a number (-0.5) or a white box is partially covering it. **\***
- Avoid rainbow color palette for panels (f) and (g), when the figure is printed in grey both ends of the scale has the same color. **\***

Fig. 3:

- y-labels for panels (b) and (c) missing **\***
- Note that the histograms in panel (b) reflect a composite of two quite different responses before and after the Quaternary. I think this makes the blue/desensitized curve look 'artificially' unimodal because of the remarkable stability prior to the Quaternary, while the Quaternary period shows a bimodality very similar to the red/sensitized scenario but with a smaller amplitude.

***This observation is entirely correct that the ice volume PDFs are not stationary and are different for different integration time periods. However, the blue (desensitized) ice volume PDF is unimodal for both the full time period and for the Pleistocene alone (see frequency distributions plotted by time period below), so this specific concern is not supported.***

[Figure]

*More broadly, the above plots highlight that the bimodality of the red (sensitized) PDF for the full model run is, in part, due to the tendency of the model to occupy a "small" end-member state during the early part of the model run (before 2.6 Ma) and a "large" end-member state during the later part of the model run (after 2.6 Ma). Thus, if a short period at the beginning of the model run were chosen, the sensitized ice volume PDF would be unimodal at a "small" state with a long tail to "large" states, and if a short period at the end of the run were chosen, it would be unimodal at a "large" state with a long tail toward "small" states. The important thing is that this behaviour clearly highlights the tendency of the sensitized model to occupy end-member states, and the tendency of the desensitized model to occupy intermediate states. In other words, we think this property of the model ice volume comparison is a feature, not a bug.*

- Panels (d) and (e): why did you pick these two time slices that both predate the max time (5 Ma) shown in panel a? Are these (7.68 Ma, 9.42 Ma) even within the modelled time range?

*This is a labeling error - they should be in ka (768 and 940 ka) rather than Ma. We apologize for this messy error and will correct it in the revisions.*

- Check formatting of 'd18O' in caption *

Fig. 4:

- Panel (a): you state in the caption that "The sensitized model (red) displays larger variation in ice thickness and is more likely to occupy extreme values, whereas the desensitized model (blue) is more likely to occupy intermediate values". To me it looks like the truth of this statement depends on what period you look at. For the last million years for example (most important for cosmogenic nuclides) both curves appear to occupy extreme values most of the time. Between 1-3 Ma, the blue curve seems to stabilize at an intermediate thick stage before thinning again in most glacial cycles, but the transition between stages still appears to be rapid. I wonder how sensitive the distinction between curves in (b) and (c) is to the choice of timescale.

*Yes, as described in our response above (regarding Fig 3), these blue/red (desensitized/sensitized) curves are different depending on the timescale. The purpose of this figure is to demonstrate that ice thickness patterns at this site generally follow the same behavior as the continent-wide ice volume curves in Fig 3: ice thicknesses tend to occupy extreme end states in the sensitized model versus intermediate states in the desensitized model. We show the dependence of the ice thickness CDFs for this site on the integration time period later in the manuscript, when we compare them to observational data at this site in Fig. 9 and section 6.3 of the original draft.*

- Caption: you refer to grey shading in (b) which is not there. *
- You refer to dashed black line in (b), which is really shown in (c). You appear to have two definitions for this line, see two last sentences. It is not obvious to me what you mean by "modern ice thickness… is chosen to approximately align the range of ice

thickness in the model simulation with that inferred from geologic evidence" (last sentence). How can you choose a modern ice thickness?

*This is an important question, and is discussed in detail in lines 386-404 of the original manuscript. We will direct the reader to this discussion section in a revised manuscript.*

Fig. 5:

- I expected a brief explanation in the caption of the difference between the top and bottom scenario. I can see that D varies between the two but consider spelling out why these two examples have been chosen.

*These examples are intended to provide a visual representation of how we calculate the metric that we introduce in this manuscript (the CDF difference metric 'D'). The upper column denotes a scenario that would result in a higher value of D, and the lower column shows a small D. A revised manuscript will modify the caption with the underlined addition: "Method of quantifying difference between model ice thickness CDFs. Upper panels represent a hypothetical site with large differences between sensitized and desensitized models, and lower panels represent a hypothetical site with similar ice thickness behavior between models. Red and blue curves exemplify output from desensitized (blue) and sensitized (red) ice sheet model runs, displayed as histograms (left) and CDFs (right)."*

Fig. 7:

- The green dots represent sites with >1 Ma histories, is that also the case for the blue dots?

*Some blue dots denote sites with exposure ages > 1 Ma, but others are representative rather than exact locations, as described in the caption. We propose the underlined addition to this caption in a revised manuscript: "The green dots are locations where cosmogenic-nuclide data from bedrock at interior nunataks indicate exposure histories longer than 1 Ma... The surrounding plots (a-l) display ice thickness CDFs… at selected sites (azure blue dots) where some cosmogenic-nuclide data with ages > 1 Ma exist. The upper and lower Lambert Glacier, Beardmore Glacier, and Byrd Glacier (l) sites are representative rather than exact data locations, because the coarse resolution of the model means that existing exposure age datasets collected adjacent to these glaciers do not fall into the model grid cell corresponding to the glacier location."*

- Label for colorbar is missing *

Fig. 8

- What nuclides are these apparent ages calculated from? Apparent ages depend in part on the half-life of the measured nuclide (e.g., 14C vs 21Ne in the same sample could yield wildly different app. ages). Are there any patterns in what nuclides are measured at different elevations/distances from coast? I would guess not but this may be worth addressing if you are mixing ages derived from nuclides with different half-lives.

*The apparent ages are computed from a variety of nuclides, mainly $^{10}$Be and $^{21}$Ne. Although the reviewer is correct in pointing out that only certain nuclides are capable of indicating very old apparent ages, this issue is not a critical element in the figure, because the figure is designed to make a distinction between exposure ages < 50 ka and exposure ages > 1 Ma and this distinction does not correspond with the saturation time of any particular nuclide. The distinction could be equally well observed in $^{3}$He, $^{10}$Be, $^{21}$Ne, or $^{26}$Al data. The exception, as pointed out by the reviewer, is that $^{14}$C data cannot indicate apparent exposure ages greater than about 30 ka, but $^{14}$C data are a small minority (5%) of the Antarctic exposure-age data set. For this reason, we kept the figure as simple as possible by not indicating which nuclide was measured. Certainly this is an interesting data set that is not completely explored by this relatively simple figure, and a more elaborate analysis might give more detailed insight into the geographic patterns of weathering and erosion in Antarctica beyond the first-order observation that surfaces higher and farther from the coast are more stable, but that analysis would be a different paper.*

Figs. 9-13

- Would it be worth explaining the double y-axes and specifying what 'h' is? *
- The legends vary between the different figures – I suggest you add nuclide name to all (missing in fig. 9 and 12). *
- You state in the paper that "26Al data can provide no information about events prior to ~3 Ma", so why did you choose to show 26Al/10Be and 26Al/21Ne ratios in the left panels (5 Ma to present)?

*We decided to show all the available data in all figures simply so that all the data would always be there for the reader to see, and not cause confusion by conflicting with other publications of the same dataset. Certainly this is an editorial decision, but we think it's a better approach than selectively editing data in a complicated way.*

Fig. 14

- There is a discrepancy between white and green dots in legend and caption. Legend has white dots as 'Any age >1 Ma' whereas caption describes white dots as 'long-exposed bedrock surfaces are not likely to exist'. Conversely green dots are 'All ages < 50 ka' in figure legend, but caption has them as 'long-exposed bedrock surfaces are likely to exist' and 'ages >1 Ma have been observed'. *
- I find it hard to distinguish the white dots on the inset panels, can they be made bigger? *

Manuscript

- l. 52 and 55: Since subglacial data has been gathered in recent years, perhaps state that this type of data is (yet) too sparse rather than saying that 'it is not possible at the moment' and 'In contrast to subglacial basins, it is possible…'. *

- l. 121: patterns in plural **\***
- l. 213-217: The described difference between the red and blue curve is not representative for the last 1 Ma. For this period, the blue curve also appears to switch abruptly between states, although the endmembers are closer together.

*As with Fig. 2, "steep" is relative and the blue (desensitized) curve does indeed switch between states slightly less abruptly than the red curve, when we expand the time axis enough in to see (zooming in to the last few glacial cycles in Fig 3a, which is what this text describes):*

[Figure]

*Although this detail is important, we feel that it is more important to show the full model run in Fig. 3a, and that the histogram in Fig 3b adequately represents the unimodal behaviour of the blue desensitized curve (tendency towards occupying intermediate values).*

> As cosmogenic nuclides are increasingly sensitive to the most recent period due to decay (and erosion, although not considered here), I think you should comment on why that is the case and whether it has an impact on your interpretations.

*We plan to clarify the text on lines 217-220 regarding the differences in ice behavior through time: "...the desensitized ice sheet is normally distributed (more frequently has an intermediate value) whereas the sensitized ice sheet is bimodally distributed (more frequently occupies extreme maximum or minimum configurations), although the details of this frequency behaviour depend on the time period of interest."*

- l. 240-241: You state that 'the desensitized model is more likely to occupy intermediate values near 1200 m'', however, it looks to me like the blue curve spends relatively little time near 1200 m within the last 1-2 Ma.

*The original text states: "...the sensitized model is more likely to occupy minimum (ca. 1000 m 240 for this example) or maximum (ca. 1500 m) values, whereas the desensitized model is more likely to occupy intermediate values near 1200 m." The 1200m value is an*

*average model thickness across the entire desensitized model run; however, a revised manuscript will remove the actual elevation values since this is not necessary to support the point of the sentence, and may confuse readers as evidenced here.*

- l. 313: should this only refer to Lower Beardmore (Fig. 7i) since Upper Beardmore fail this criterion? *
- l. 370: Would it be worth also mentioning that the exposed bedrock would need to contain minerals where production rates are well-calibrated for nuclides with half-lives that cover the relevant timescales?

*Yes, this is a necessary consideration for moving forward with model/data comparison, but it's mostly not of critical importance -- as a practical matter, there are almost no lithologies outcropping in Antarctica that don't permit measurements of at least a couple of useful nuclides.*

- l. 415: Do you need a reference for the ICE-D:Antarctica database according to journal guidelines?

*We believe this reference should be acceptable, but will defer this issue to copy editors.*

- l. 425-430: consider citing data references in this and the following sections, I see them in the figure captions, but not in the main text. *
- l. 466: specify 40-km resolution model *
- l. 498: change 'there do exist rock outcrops' to 'rock outcrops do exist' *
- The approach in this paper regarding constraining long-term ice-sheet cover based on an elevation transects of cosmogenic nuclides seems comparable to the one in "Jones, R. S., Norton, K. P., Mackintosh, A. N., Anderson, J. T. H., Kubik, P., Vockenhuber, C., ... & McKay, R. (2017). Cosmogenic nuclides constrain surface fluctuations of an East Antarctic outlet glacier since the Pliocene. Earth and Planetary Science Letters, 480, 75-86." Would it be worth a citation?

*This is definitely a relevant citation for our approach - this omission will be rectified in a revised manuscript.*

- Consider spelling out MPWP since you only use the abbreviation a few times. *

---

## Author Comment (AC2)

**Response to review 2 of 'Cosmogenic-nuclide data from Antarctic nunataks can constrain past ice sheet sensitivity to marine ice margin instabilities' by Halberstadt et al.**

The paper investigates the use of cosmogenic nuclide measurements from nunataks across the interior of Antarctica to evaluate the performance of ice sheet model simulations over past time periods. Under certain assumptions and requirements, the inferred frequency at which the sample sites have been covered by ice can be used as a proxy to determine a cumulative distribution function for ice sheet thickness in the surroundings of the nunatak. Uncertainties aside, the same quantity can be derived from an ice sheet simulation for that location, and thus a direct comparison between model and data is possible. In addition, the authors propose a metric based on the model version of this function to identify regions where different model realisations exhibit large discrepancies in their integrated behaviour. According to the authors, future measurements over some of these regions could provide valuable benchmarking capabilities for ice sheet models used in both paleo reconstructions and future projections. In my opinion, model-data integration is a topic in ice sheet modelling that has been undeservedly left as a secondary problem due to computational limitations that, so far, have made it difficult to create bridges between the field and modelling communities. As shown by the authors, the method explored in this study is a valuable approach that can, in principle, be used to evaluate virtually any modern paleo ice-sheet set-up. As more precise and abundant field data are gathered, and as the capabilities of state-of-the-art ice-sheet models increase, the applicability of this method will likely improve. As such, the core of the study is a nice addition to the scientific literature and, in this sense, I support its publication in TC. However, there are several points which I found problematic in the way the authors chose to showcase the applicability and significance of the method, particularly within the modelling set-up. I think that these choices do affect the conclusions, and that confirmation that this is not the case is needed before acceptance of the manuscript. I will describe these points as part of my general comments below, and if necessary elaborate on them in the specific comments after that.

*We thank Dr. Bernales for the thoughtful and insightful analysis. We directly respond to each comment in detail below, and describe throughout how we plan to revise the manuscript to address these issues. We propose three main revisions: Clarify our aims and approach (Revision 1), ensure that we properly describe our ocean forcing method as an external forcing rather than an internal nonlinearity (Revision 2), and clarify model details in an added appendix (Revision 3).*

General comments:

1. As the authors acknowledge ("Criterion 1" in Discussion), the model simulations need to have sufficiently different CDFs for the method to be usable as a constraining tool. The authors then generate these two sufficiently different model realisations by including/excluding a representation of the ice shelf hydrofracturing (HF) and marine ice cliff instability (MICI) processes, and by simultaneously scaling up/down the time-dependent ocean temperature shift that the model uses to derive melt rates at the base of ice shelves. To clarify my concern, and to prevent specifics worries being diluted in a single comment/response, I will divide it into several

smaller points:

*We respond briefly to points A-G in line (to prevent dilution of specific worries), followed by a full response to comment 1 at the end.*

A.  The authors put both HF and MICI in the same category as the stronger/weaker ocean forcing, and name them all "marine ice margin instabilities". I agree with HF and MICI, as these processes are inherent to ice dynamics, i.e., the modelled ice is reacting (or not; through crevasses and calving) in some consistent way to some time-varying external boundary conditions (e.g. surface temperatures, meltwater ponds, accumulation). If a fixed change in the external forcing is applied, both model realisations will react very differently, with the one including both HF and MICI potentially generating positive feedbacks that can trigger significant ice sheet retreat (e.g. [1]). In other words, a model with HF and MICI *is* more sensitive to a given change in the external forcing.

*Agreed. In our research design, we varied HF and MICI parameters in order to produce the largest possible variation in behaviour between ice sheet end-members.*

B.  However, applying different ocean temperature shifts is no more than simply applying different changes in the external forcing, not inherent to ice dynamics: apply the same change in ocean temperatures (the external condition) to both models, and the resulting thinning/thickening of ice will be the same. Here, ocean temperature reflects an uncertainty not in model sensitivity, but in the external forcing. In other words, both models are *equally* sensitive to a given change in ocean temperature, and thus equally (un)stable and their responses equally (non-)linear. In this regard, the study is not constraining marine ice instabilities, but simply stronger vs. weaker forcing conditions.

*Agreed. We have improperly implied that the ocean forcing is an internal nonlinearity.  As the reviewer highlights here, we combine two different strategies -- nonlinear processes and forcing changes -- to achieve our aim of producing the most extreme end-member ice sheet behaviour (and thus elicit the largest difference between model simulations). Therefore, we clarify our approach (Revision 1) and correct our language regarding ocean forcing and end-member parameter choices throughout the paper (Revision 2). Revisions are described more fully below.*

C.  A way to include different sensitivities to a given ocean temperature change would be to use two distinct ocean melt parameterisations, e.g. one where total melt is linearly proportional to ocean thermal forcing and another that is quadratic. Those two will react very differently to, say, a 1 K change in local ocean temperatures. Another would be to test melt under partially floating grounding line cells vs not doing it (e.g. [2]), which as a bonus is, in my opinion, equally controversial as HF and MICI. Both options work directly on an inherent ice dynamics quantity, namely dH/dt, instead of the external forcing. That I would consider closer to a "marine ice margin instability", together with HF and MICI.

*This is an excellent suggestion to guide the next step in this research trajectory. After demonstrating that large model behaviour differences can indeed be resolved using exposure age data, as we show here, the next step is to directly test model parameterizations using more realistic (rather than end-member) simulations, and this controversial ocean melt parameterization is a good candidate for this kind of paleo-validation. (However, we also note that multiple new datasets will also need to be collected in the future that are sufficient for this use, in addition to the one existing Pirrit Hills record, in order to move forward and confidently constrain model parameterizations).*

D.  Why such a strong difference in the two end-member simulations? HF and MICI alone (albeit only in tandem) were enough to generate a huge difference for the Pliocene in [1]. Adding on top of that two full degrees Celsius to each side of the ocean temperature shift for the sensitised model feels overkill. And that's not even considering the extra warming added during interglacials to the Amundsen Sea area.

*Our approach in this work was to generate extreme end-member model simulations that highlight the maximum possible variation in ice-sheet behaviour. We follow ref [1] to choose parameters associated with extreme Pliocene ice sheet retreat; in addition to the HF and MICI parameterizations, they applied a +2°C ocean temperature shift (along with additional bias correction added to the Amundsen Sea). We chose to ramp ocean temperatures by a maximum value of +3°C to ensure a maximum ice sheet response in accordance with our aims (for glacial periods, we chose a -3°C shift based on Liu et al. (2009) temperatures at LGM, as noted in the manuscript L186, along with sensitivity testing of this ocean temperature shift parameterization that produced a decent LGM and modern configuration). Note that these maximum temperature shifts are weighted based on the global oxygen isotope record and insolation, so the full ocean forcing is only applied fully to the model when past d18O/insolation values reached their extremes.*

I imagine that the method would struggle with less extreme differences, i.e. with fulfilling Criterion 1. What happens when only HF and MICI are used, but not different ocean forcings? Are the resulting CDF's still different enough to make the method usable? Could it be that the sensitised CDF fit is poorer than the desensitised one because of this extreme ocean influence? See also my specific comment L38-39 below.

*We were also curious about these questions, and ran some additional model simulations to investigate (see response to E/F).*

E.  I have the inkling that the ocean forcing is dominating the apparent bimodality of the sensitised CDF, which for me would be problematic, since i) I consider it as external forcing, not as an instability, and ii) it is, as far as I can tell from the manuscript, heavily parameterised. If this is the case, then the method is not constraining marine ice margin instabilities, but simply which parameterised forcing is less unrealistic.

F. A way to confirm this would be to run a third simulation with HF/MICI, but with the same ocean as the desensitised run (i.e. [-1 1]), and a fourth simulation with the ocean full temperature difference applied (i.e. [-3 3]), but with no HF/MICI. That would provide a clearer way of interpreting the model results, and –more importantly-- an assessment of the applicability of the method.

*Below we plot new simulations spanning the last 130ka that are sensitized versus desensitized to HF/MICI (ice margin instabilities) under high and low ocean forcings (±3°C or ±1°C). The reviewer is entirely correct that the ocean forcing significantly impacts the frequency distribution (histogram shape) of our 'sensitized' and 'desensitized' simulations. We also note that the time scale of consideration also affects the shape of the histogram (see response to Reviewer 1), as do other parameter choices. These plots highlight that the ice sheet model is inherently non-linear, even without the HF/MICI instability parameterizations, if a big enough external forcing is applied (e.g., ocean temperatures). Basically, as shown here, we get more linear ice sheet behaviour with lower sensitivity to HF/MICI and/or lower-amplitude ocean forcing. We get more non-linear ice sheet behaviour with higher sensitivity to HF/MICI and/or higher-amplitude ocean forcing. These plots simply show that the higher ocean temperature shift forces the model into non-linearity/instability more often.*

[Figure]

[Figure]

*We interpret this additional analysis as a somewhat interesting property of our models, but it does not impact our conclusions or detract from the value of this work (see full response below). Our goal in this work was to elicit the largest possible divergent response in our ice model simulations, which we have done by combining parameterized sensitivity to instability mechanisms as well as the amplitude of the ocean scaling (external forcing).*

*We disagree that this method only constrains "which parameterized forcing is less unrealistic"; our goal here is not to produce any realistic model forcing at this stage, but simply to present a novel method of using long-term exposure ages to constrain model simulations and demonstrate that this method can indeed be used to distinguish between end-member simulations.*

G.  If my concerns are confirmed, most of these issues can be alleviated a bit by toning down the title and avoiding specific claims, e.g., "ice sheet sensitivity to marine ice margin instabilities" -> "strong vs weak ice-sheet retreat and advance histories".

*In summary, we believe that the reviewer concerns throughout comment #1 can be alleviated with two proposed groups of revisions:*

*Revision 1: Clarify our aims and approach. Our aim in this work was to create extreme end-member model simulations that highlight the maximum difference in ice-sheet behaviour, in order to see if exposure age data can resolve large differences in model simulations (and we find that they can!). We did not intend to present these end-member simulations as fully realistic; the next step in this research trajectory is therefore to produce an ensemble of more realistic simulations and then try to use these data to actually say something about what parameterizations are appropriate for past (and future) warm periods. To address this subsequent key question, the ensuing scope of work would need be significantly larger, including: (a) multiple new elevation transects of exposure ages; and (b) more detailed and refined analysis of model frequency behaviour, as the reviewer brings up here, as well as considerations regarding geographical fingerprinting of different mechanisms/feedbacks as well as the time interval of integration (as Reviewer 1 notes).*

*Suggested edits include L38-43: "The goal of this paper is to explore how to use geologic data from the Antarctic continent to differentiate between ice sheet model simulations with end-member instability behaviour (e.g., Fig. 1b versus 1c). We aim to elicit the largest possible variation in model ice sheet behaviour in order to test if this difference is resolvable using cosmogenic nuclide data. We describe end-member simulations as 'sensitized' or 'desensitized' models based on the idea that stronger positive feedbacks in the form of marine ice instabilities result in model predictions that are more nonlinear, that is, more "sensitive," with respect to the forcing. Specifically, we investigate the sensitivity of ice sheets to marine ice cliff instability under stronger and weaker ocean temperature forcing."*
*L56: "Therefore, our goal is to quantify if, where, and when sensitized and desensitized model simulations make different ice cover history predictions for Antarctic outcrops where corresponding geologic data already exist or could be collected. At these locations, we can compare model predictions to geologic data as a means of gaining insight into past ice sheet behaviour. This methodology therefore can be applied to ensembles of simulations with more realistic and varied parametrizations to test which model realization most accurately represents the true ice sheet response to warm climates."*

*Revision 2: properly describe our ocean forcing method as an external forcing rather than an internal nonlinearity. We will clarify that we have mixed our parameterization of nonlinear mechanisms with the amplitude of ocean forcing to elicit the largest possible*

*model response, and thus we will ensure that we no longer refer to our varied parameterizations as simply 'marine instabilities'.*

*We will clarify our approach regarding these parameter choices in the L38-43 paragraph described above, as well as add additional discussion on L137 (after introducing the instability mechanisms that characterize our 'sensitized' and 'desensitized' end-member models): "To elicit the largest possible difference between these end-member simulations, we further enhance the ice sheet instability mechanisms in the sensitized model with a stronger ocean temperature forcing, whereas the desensitized model experiences weaker ocean forcing".*

*We will also reword the relevant portions of the manuscript where we refer to 'marine ice instabilities' to more accurately reflect our approach. For example, L45: "The critical difference between sensitized ice sheet models (with strong marine ice margin instabilities and a strong ocean temperature forcing) and desensitized models (with weak instabilities and weak ocean forcing) is the extent of deglaciation of marine basins". This will entail text edits throughout the manuscript, including the abstract.*

*The title will be altered to: "Cosmogenic-nuclide data from Antarctic nunataks can constrain past ice sheet instabilities". We realize that this still invokes the presence of instabilities, but we feel that this title still accurately reflects our main point (which is that this methodology can be applied to model simulations to distinguish between instability parameterizations, and this is how we hope that the wider glaciological community will utilize this method in the future). While it is it true that our ocean forcing parameterization controls only the amplitude of the external forcing, it represents a non-linear instability in ice-ocean interactions which is not accounted for in the model - so our approach ultimately is geared towards investigating the potential signature of glacial behaviour under strong or weak non-linear instabilities.*

*Finally, we address the reviewer concern that "these choices affect the conclusions." Indeed, our parameter choices certainly affect the frequency behavior of our end-member simulations, but this was our intention (which we have now clarified more fully), and does not impact our conclusions, namely, that large differences in model simulations are resolvable using geologic datasets.*

2. While reading the paper, it is not fully clear to me what exact version of the PSU3D ice sheet model the authors used, and therefore how different processes are exactly represented in the study. The authors first introduce (L28) the model with some results from the 2015 version [1], but then cite (L169) the 2021 version [3]. Then, they mention (L172) that the approach in the 2009 version [4] is needed for the external forcing, for which I assume they use a modification of it since modern data sets of surface air temperature, precipitation, and subsurface ocean temperature are here mentioned (none of which is part of [4]), but this modified approach is only modestly described, and without any formulae (e.g. L173 "scaled

based on a combination of factors"). Then, an additional parameterisation is mentioned for ocean temperature shifts (L175). Finally, L191 mentions an additional modification for the Amundsen Sea, only for the sensitised model, and only during interglacials:

A.  In this version of the manuscript, it is not possible to get a clear idea of the forcing implemented. An appendix is needed where this and every other change with respect to [3] is provided clear and transparently. These details are important since the conclusions might depend on them (see my general comment 1).

*Thank you for this suggestion; we will add an appendix outlining more details of our model components and application here (Revision 3). This appendix will include:*
-   *Description of the long-term forcing scheme we applied here, which is an update of the Pollard & DeConto 2009 [4] weighting scheme. The ocean temperature ramped scaling follows the weighting scheme as described in Pollard & DeConto, 2009, and will be re-described here.*
-   *Explicit citations for input data fields - atmosphere temperature and precipitation: SeaRise climatology (Le Brocq et al., 2010); ocean temperature: World Ocean Atlas (Levitus et al., 2012; Locarnini et al., 2010) at 400m water depth.*
-   *Further description of the additional temperature shift added to the Amundsen and Bellingshausen Seas during interglacial time periods in our sensitized simulation. This technique was introduced as a bias correction by DeConto & Pollard (2016), to bring modern modeled ocean melt rates closer to observations of recent warming in this region (e.g., Schmidtko et al., 2014). DeConto & Pollard (2016) originally applied a 3°C temperature shift; here we use 1.5°C following DeConto et al. (2021). However, DeConto & Pollard (2016) note that this effect has no impact beyond a few thousand years. This temperature addition is applied in our transient simulation as a ramped temperature shift similar to the ocean temperature scaling that will be described more fully also in the appendix.*

B.  With the above modifications, this is a different model and thus it does not make much sense to reference previous studies in terms of specific model behaviour. The introduction mentions the difference between models with and without HF and MICI (L30 onwards), which sets the ground for the rest of the paper in terms of instabilities, retreats, sea level contributions, and the inability of previous attempts at discerning between such models (L35). Finally, the goal of the paper and the sensitised vs. desensitised models are introduced (L38-42) within this context. This whole introduction heavily uses Fig 1, which represents data not from this study, but from [1]. After the significant work that running 5 Myr simulations I assume implies, which contain almost all periods covered by [1], [3], and [4], and the fact that this is a different model version, I expect the authors to illustrate their points, and in particular Fig 1, using the output from the model version used in this study. This will be a great chance to showcase the performance of both simulations for not only the Pliocene and present-day, but other crucial benchmark periods, such as 125 ka and 21 ka. See my specific comment L170-171 below.

*We feel that the current Fig 1 (based on previous studies) adequately fills its intended role, which is an introductory figure that uses existing knowledge to motivate the current study, rather than presenting results or prompting discussion. A secondary reason to keep Fig. 1 simple (e.g., based on existing work) is that our intention in this work is not to produce realistic ice-sheet configurations but rather to produce end-member ice sheet behaviour.*

*See response to specific comment L170-171 for snapshots of our sensitized and desensitized model ice sheet configurations during these time periods; we will also consider adding these snapshots to the model methods appendix (Revision 3).*

3. Missing critical reference for the insensitivity of results to model resolution (L171-172). I assume you want to cite one of the model intercomparison initiatives where results seem to suggest that coarse grid resolution models can bypass the 100s-of-meters grounding line resolution requirements, when using e.g. Schoof's flux parameterisation or a sub-grid interpolation of basal friction at the grounding line (I am also assuming you are using the former). However, those intercomparison exercises usually employ idealised geometries that are far from the steep topographic features found in Antarctica, especially around nunataks:

A. I'd like to see some kind of indication that the results presented in this study and/or the version of the model used by the authors are/is resolution-independent under a similar Antarctic set-up.

B. If such an indication does not currently exist: I understand that 5 Myr is in general computationally expensive, but comparison against sub-set periods, say, even 500 kyr at 32/20 km and 250 kyr at 16km (assuming a ~ten-fold increase in comp. time per doubling of resolution) should allow you to confirm the claim in question, at least regarding the scope, assumptions, and conclusion of this study. This is valid even when the geological data would require longer time spans: the goal is to check how different are the CDF's for limited sub- set periods between simulations at varying resolutions, and not against the data.

*Model resolution insensitivity has been demonstrated through idealized model intercomparisons (e.g., Pattyn et al., 2013, doi:10.3189/2013JoG12J129), and has also been documented for transient continental-scale runs, as shown in a few sub-sections of different papers:*
- *Pollard et al., 2015, Supplementary Information S5: "To evaluate the effect of grid size, we repeated the modern and warm-climate simulations (shown above at 10 km resolution) at resolutions of 40 and 20 km for continental Antarctica, and at 5 km for nested domains over the Wilkes, Aurora and Recovery-Slessor-Bailey basins. As shown in Fig. S5a (top row), the modern ice distributions are very close to each other at all resolutions, as expected from previous modern tests (Pollard and DeConto, 2009, 2012a)."*
- *DeConto et al., 2021, Extended Data Fig 5g, shows systematically varied resolutions in nested simulations of Thwaites basin retreat resulted in very little*

*dependence on resolution.*

*Previous (unpublished) experimentation throughout the years has continued to uphold model insensitivity to grid resolution in many offline long-term continental runs. We expect this to intrinsically be the case based on the model design, specifically the Schoof grounding line flux parameterization. This grounding line treatment avoids the transitional grounding-zone dynamics that require locally very fine grid resolution, and depends only on grounding line depth which is sub-grid interpolated between grid points and so doesn't change much with resolution (because the sub-grid interpolation is accurate to the limit of the resolved bathymetry). This is also true for diagnosed stresses and ice-cliff failure rates, which should make the model largely independent of grid resolution.*

Specific comments:

*We respond in detail to substantive specific comments below, with suggested revisions to address them. The review also contained a number of technical corrections and suggestions for clarification; where we give no specific response to one of these technical corrections or clarifications, we agree with it and will incorporate it in the revision.*

L26 – Could you please elaborate on what you mean by "ambiguous", and add it to the text? The word is used a few times throughout the manuscript, but the only time it is elaborated is in L451, where data are not dense enough (that would be insufficient) or taken from an approach-irrelevant elevation (that'd be uninformative). That is not a problem of the data, though.

L28 – Maybe a matter of opinion, but I would argue that is the present climate (~415 ppmv, ~1 ºC over Pre-Industrial) state the one approaching the peak mPWP states (~350 - ~450 ppmv, ~2.0 - 4.0 ºC above Pre-Industrial, inc. higher sea level), at least when acknowledging the poor paleorecord, uncertainties, and biases. Perhaps removing this sense of direction altogether? Refs [5] and [6] below (and therein) contain classic, neutral examples of this comparison.

L30 – This would be a good place to introduce the concepts of ice shelf HF and MICI, after "… ice margins (Fig.1)" and before "Model runs…". These concepts are used extensively in this study, and it would be nice to minimally describe them here. Otherwise, the reader is left hanging until L43, where "ocean temperature" is added to the mix (see my general comment 1), only HF is mentioned (not described), and where my brain predicted MICI to be also mentioned (but it was not). This is because [1] show that, at least in their highly parameterised and somewhat speculative (their words) simulations both HF and MICI are needed in tandem to get a retreat similar to that shown in Fig 1c. The reader is then left hanging again until some more info is given in section 3.1, which was in my experience distracting.

L38-39, L56 – This connects to my general comment 1. If the aim of the paper is to explore how to differentiate between margin instabilities, and Fig 1b and 1c are given as examples, then by using buffed and nerfed ocean shifts compared to the homogeneous +2 K that [1] used for their Pliocene, then the answer to the interesting question is obscured: "If we adapted both simulations F1b and F1c in [1] to a 5 Myr run (i.e. turning [1] into [4]), would we be able to discern between them using geological data and CDF's?" That's a beautiful question, but by modifying the ocean the answer is biased.

*See response to comment #1 - but yes, we hope that future work will now be able to apply the methodology described here in order to answer these more involved questions!*

L71 – Section 2 was a pleasure to read.

L129 – There it is: "given the same external temperature and accumulation forcing". I fully agree.

L135 – "By a more linear", perhaps? There and elsewhere

L136 – endmember -> end-member (e.g. L124)

L138 – "Gray bars", maybe?

L148, L161 – Here MICI is mentioned, but not ice shelf HF. See my comments above on "Line 31". Consistency is needed on how you define and utilise these terms throughout the manuscript.

L150 – See my general comment 1, and specific comment L39-39, L56 above. Marine ice sheet instability (MISI) is not something that you can (or rather should) "turn on or off" in ice sheet simulations; it is a consequence of a retreating grounding line on a retrograde bedrock slope. If you try to supress it by using colder ocean temperatures during warm periods (colder than the +2 K needed and justified by [1] themselves), and then concluding that geological data support this, that's a very bold statement.

L162-165 – That's a long parenthesis. I'd suggest let it be its own full sentence.

L170-171 – See my general comment 3.

L172-180 – See my general comment 2A. It would be much clearer to add an appendix with the exact formulations (i.e. in equation form) mentioned/used in this study.

L173 – Which modern forcing data sets? Model-based (e.g. RACMO, MAR) or observational (e.g. WOA)? See my general comment 2.

L181 – Do you limit the lower shift in ocean temperatures based on the local pressure-freezing point of sea-water? Otherwise (assuming no supercooled water processes are

resolved by the model) you could be assuming a below-freezing ocean in many areas during glacials. That would create mass "out of thin water", artificially making your ice sheet advance and your ice shelves proliferate.

*Yes, we follow Martin et al., 2011 (doi:10.5194/tc-5-727-2011) to scale temperatures under the ice shelf, which takes into account the pressure melting temperature (although our formulation uses a quadratic dependence of melt relative to temperature above the melt point). This description (along with a reference to Pollard et al., 2012, who describe our implementation) will be added to an appendix (Revision 3).*

L184 -- Same as specific comment L162.

L186 – Missing parenthesis at the end of the TraCE reference

L187 – I guess you mean "desensitized" here.

L189-190 – This is key to avoid model biases contaminating your conclusions, and it surely needs a figure supporting this claim. The easiest would be to show Pliocene (classic), 125 ka (recent interglacial example), 21 ka (recent glacial example), and transiently reached 0ka, for both simulations. A figure like that would be much more informative than current Fig. 1. See my general comment 2B.

*This particular sentence ("Both the sensitized and desensitized ocean temperature scaling parameterizations yield reasonable glacial maximum extents at ~20-15 ka with subsequent retreat to approximately modern configurations by 0 ka") describes the 'sanity checks' we did to ensure that our model parameterizations, while intended to be representative of end-member behaviour rather than ultra-realistic, were indeed reasonable for known time periods (LGM and modern).*

*Our intention with Fig 1 was to motivate the current study using existing knowledge (see response to comment 2B). We did not create a figure specifically to highlight model behaviour during these key time periods because our intention in this work is not to produce realistic ice-sheet configurations but rather to produce end-member ice sheet behaviour. However, ice-sheet configurations during these times can be useful to highlight differences in end-member behaviour; we show snapshots of these times below (grounded ice only):*

*"MPWP" (a sample interglacial at model time = 3850 ka):*

[Figure]

*Last interglacial (model time = 120 ka):*

LIG:

[Figure]

*Last glacial maximum (model time = 20 ka):*

[Figure]

*Modern:*

[Figure]

*Note that the sensitized model produces a partially collapsed West Antarctic Ice Sheet for the modern time period. We don't interpret this as a 'result' indicating that the sensitized parameterizations are less appropriate, given that our goal was to produce end-member behaviour. Model instability of the WAIS under the modern climate is a well-known feature of ice-sheet modeling, even without HF/MICI instability mechanisms. This simply indicates that the sensitized simulation is responding to modern climate a few kyr faster than the desensitized simulation, which is not relevant to the time scales we consider here.*

L191-196 – See my general comment 1. As it is surely evident by now, I am not easy with forcing the models by using different ocean temperature shifts and calling that a margin instability. Here, an even stronger difference is applied between the two simulations. What happens without this 1.5 K extra increase? How is the factor applied only during interglacials: as a shock once a time-point is reached, or smoothly as the interglacial is approached? Why is that trend not already in the modern ocean data set used as forcing? Even within quotes, a +1.5 K warming is hardly noise for a simulation that is limited to a modest +1 K above modern.

*See response to comment #1 regarding our intent to better describe the ocean temperature shift as an external forcing rather than an internal nonlinearity. The 1.5°C increase in the Amundsen Sea region is an established protocol for Pliocene simulations (ref[1] in this review) and here is applied smoothly (ramped) as an interglacial is approached, following ref[3]. This is intended as a bias correction to the Levitus 'modern' ocean temperature field used as model input, based on ocean temperature measurements in this region collected more recently.*

Lines 197-208 – Again, a summary of the mathematical expressions of the code for key processes discussed in the paper (e.g. atmosphere/ocean forcing and how it evolves, ice shelf HF, and MICI, …; i.e. all about how both simulations differ) is a currently missing appendix. Otherwise, the reader is forced to navigate through the last 10 years of PSU3D development without a clear idea of the precise version of the model used in this study.

L220-222 -- See my general comment 1. I would argue that using different ocean temperatures in your simulations qualifies as "a property of the forcing dataset", which could be in fact playing a primary control on your characteristic model behaviour. By the way, if I understand your method, d18O is not the only forcing used, but insolation and CO2 are involved as well. It'd be nice to have frequency distributions for those as well, and a distribution of the final weight used in the forcing parameterisation.

*Yes, the weighting is proportional to d18O as well as Southern Hemisphere summer insolation (but not CO2 explicitly; we assume that the effect of CO2 is included in the dO18 variations, because glacial/interglacial CO2 cycles in ice cores are highly correlated with d18O). This will be described more fully in an appendix that provides*

*more details of our modeling approach (Revision 3). Unfortunately, we did not output the overall weight in our 5Ma simulations, but one can visualize the combination of 5Ma-0 d18O and summer insolation histograms, shown separately below:*

[Figure]

I suspect the additional scaling added to the ocean temperature shifts, which differs between the models, could be bimodal as well. In other words, a frequency distribution of the desensitised ocean temperature shift would be more centred around 0 (by definition, since it is capped between -1 and 1 K) than the sensitised one, which would sit more often in either of the extremes. In that case, I would think that the bimodality of the sensitised simulation is strongly correlated to the ocean temperature applied, which in this case I consider part of the forcing, rather than an ice dynamical instability represented in the model. Could you please confirm this?

*The applied ocean temperature shift is weighted by the global d18O stack (as well as summer insolation), so the only thing that changes between the sensitized and desensitized simulation is the width of the histogram (e.g., 1 or 3°C). The histograms otherwise have the same shape as the overall forcing.*

L223-228 – I am not convinced. See my general comment 1. Even if the simulations resemble the conceptual example, this could be a "self-fulfilling prophecy" case, with the external ocean forcing the main suspect.

*This comment refers to our assertion that "Our simulations of sensitized and desensitized ice sheet behaviour closely resemble the conceptual example in Section 3, but use a robust numerical model with realistic physics (Fig. 3a,b)... This confirms that our model approach has successfully promoted 'linear' vs. 'non-linear' ice-sheet behaviour by varying the parameterized ice sheet sensitivity to marine ice feedbacks [and ocean forcing]." (brackets denote proposed addition as part of Revision 2).*

*While it is true that the higher ocean forcing triggers more non-linear ice-sheet response even with lowered sensitivity to instability mechanisms (see plots and response to comment #1F), our goal was to produce the most overall non-linear vs linear ice sheet*

*behaviour by varying both model sensitivity to instability mechanisms along with the amplitude of external forcing (ocean temperature shift). We feel that our approach has indeed "successfully promoted 'linear' vs.'non-linear' ice-sheet behaviour" - Fig. 3 shows how the model d18O-based forcing (Fig. 3c, which has the same shape for both sensitized and desensitized simulations, see our response to the preceding comment) drives either a linear or non-linear model response (Fig. 3a,b).*

L242 – "desensitized" perhaps?

L261 – MICI not mentioned.

L268 – You mean Fig. 6c, maybe?

L294 – "Previously" not needed, in my opinion. Sounds to me like a past publication.

L377 – "inverted for the fraction of time ice covered". Feels like something is missing here.

L378 – What time period do these data cover? Are you comparing it to the model CDFs computed for the same period?

L392 – That's confusing. Then against what were the models calibrated?

*Again, the setup of the end-member models was to demonstrate contrasting model behaviour. Although they produce reasonable results for LGM-to-present ice sheets, they were not specifically tuned to match modern conditions.*

L400 – "differences", thus "result"

L403-404 – Bold statement. Difficult to assess without a clear quantification of the isolated impacts of a highly uncertain ocean forcing. Even if that were the case, then the desensitised simulation would need to be further evaluated against several other constraints (e.g. [7]), since it was "not benchmarked against any geological or glaciological data" [8].

*We agree, and we have therefore put as many qualifiers ("if," "imply,", etc.) in this statement as we could fit! But the overall point is important: IF this result was robust, it would be significant."*

L407, L409 – Consistency using "data set" vs "dataset". Here and elsewhere.

L417 – "Matlab code" sounds too specific, what about "approach" or "algorithm"?

*Our intent here is to indicate that exactly the same code was used, in contrast to applying the same algorithm using different code.*

L421 – "WAIS at this site" -> "this site", maybe?

L421-422 – I would say "and has, in this case, no resolving power". There is always the chance that in the future, a model will be able to better resolve this area, and perhaps generate distinct CDF's. Same with L429.

L445 – If I understand the method correctly, you should choose an integration time matching the field data temporal span, or?

***This is correct in general terms. However, in detail this calculation depends on both the half-life of the nuclide in question and the erosional history of the surface, so it is not possible to give a standard integration length for each nuclide pair. There is more detailed discussion of this issue in the Spector et al. (2020) reference.***

L453 – Although (to my surprise) it exists, "resolvably" does not seem to fit the context here.

L472 – "Collection" is singular, thus "is", perhaps?

L509/510 – Not sure that the data are used to differentiate between the two models, as the models can be separated without the data. Maybe "evaluate the performance"?

L522-526 – I think that the strength of these claims needs to be checked against my concerns above. In any case, and before such a hypothetical strong implication (L524) could be trusted, a significant amount of validation would be needed. Considerably higher resolution models, with the ability to zoom in around outcrop areas and correctly approximate the dynamics of ice in high complexity topographies, would be first. A way more realistic external forcing would be needed as well. And then, validation of the results against a myriad of other geological constraints and modern observations would be required, especially if the validation of the model has the ultimate goal of enabling a simulator of future ice sheet dynamics.

***We propose a revision of this sentence: "If replicated at multiple sites across the continent, and under a larger range of model experiments, this would be an extremely significant result, …"***

L546, L548: links seem unnecessarily complex: https://doi.org/https://doi.org/

Figure 1 – This figure can be much, much improved. As a minimum, I would expect a way to quantitatively interpret the contours and colormap, including the name of the plotted quantity (ice surface elevation I assume). Maybe a consequence of the colormap choice, but where are the ice shelves extents and their thickness in subplot a (and presumably b and c)? Modelled boundaries (grounding line and calving front) would help too. Adding the present- day observed grounding line and calving fronts in a distinctive manner, to allow for a quick evaluation of the model's ability to reproduce the present state of the ice sheet, would be useful to the reader as well. You can optionally take the opportunity to add some general Antarctic locations here as referenced in the paper (for the reader's convenience), as precise

locations are introduced only starting on Fig 7, central panel.

By the way, the caption of Fig 4 mentions that the location of Pirrit Hills is shown in "Fig. 7e" (instead of central panel), which is confusing as it seems that you are using the lowercase letters to reference both locations on a map and subplots (with the latter being wrongly referenced anyway). In any case, I'd rather have it replaced by a figure showing data from the simulations carried out in study (see my general comment 2B and specific comment L189-190).

*We will implement a colormap and caption clarification. However, the purpose of this figure is to frame the problem for a more general audience (see response to comment#2B), so we prefer to leave the figure mostly as it is currently.*

Figure 2 – Those bars are grey, in my opinion. That rectangle with the arrow is black, but I am not sure about the need for it, or its accuracy: the subs d and e do not seem to correspond. Figure needs more love: It seems to me that you forgot to correctly white- rectangle-cover the "5"s in the horizontal axis of subfigure b. "Scaled" is not capitalised in subfigures a and b; same with "time" and other words. Please make the capitalisation consistent across all figures. Style of "(f,g)" not consistent, recommend using "Desensitized … (f), while sensitized … (g)".

Subfigures f and g are counterintuitive: the linear response shows actually more retreat than the nonlinear one, which contradicts the rest of the paper. What process would cause something like that? Is the figure made up or from the simulations? Subs i and j would benefit from using the same colour scheme, preferably colour-blind friendly, to show corresponding surface levels.

*Subfigures f,g currently show the same amount of retreat in both linear and non-linear scenarios (both subfigures show the same initial and final grounding line positions); these are hypothetical demonstrations of what linear vs non-linear behaviour looks like spatially (ie. made up!). A revised figure will alter these subfigures so that the sensitized example occupies greater max and min grounding line configurations (in a colorblind-friendly color scheme), following this suggestion.*

Figure 3. What are the dates in d and e? I thought the model runs between 5 Ma and present. Are these from the current study? What is the y-axis in b and c? Some figure details need a minimal explanation in the captions, otherwise they assume the reader knows already those details.

*This is a labeling error - they should be in ka (768 and 940 ka) rather than Ma. Subs (b,c) are histograms so the y-axis will be labeled as 'Frequency'.*

Figure 4 – Photo credits? You might get some copyright warnings later.

Figure 6 – Why is D so big over the continental shelf and beyond? Does the ice reach the model domain boundaries?

*Our metric D is based on ice thickness including ice shelves, so the large spatial extent of D is due to a few time periods when extremely large ice shelves formed in the model.*

Figure 7 – Why the (l) only for Byrd in the caption?

*This can be removed.*

References:

[1]  David Pollard, Robert M. DeConto, Richard B. Alley: Potential Antarctic Ice Sheet retreat driven by hydrofracturing and ice cliff failure, Earth and Planetary Science Letters, 2015, https://doi.org/10.1016/j.epsl.2014.12.035.

[2]  Seroussi, H. and Morlighem, M.: Representation of basal melting at the grounding line in ice flow models, The Cryosphere, 12, 3085–3096, https://doi.org/10.5194/tc-12-3085-2018, 2018.

[3]  DeConto, R. M. and Pollard, D.: Contribution of Antarctica to past and future sea-level rise, Nature, 531, 591–597, https://doi.org/10.1038/nature17145, 2016.

[4]  Pollard, D. and DeConto, R. M.: Modelling West Antarctic ice sheet growth and collapse through the past five million years., Nature, 458, 329–332, https://doi.org/10.1038/nature07809, 2009

[5]  Haywood, A., Dowsett, H. & Dolan, A. Integrating geological archives and climate models for the mid-Pliocene warm period. *Nat Commun* **7**, 10646 (2016). https://doi.org/10.1038/ncomms10646

[6]  Stokes, C.R., Abram, N.J., Bentley, M.J. *et al.* Response of the East Antarctic Ice Sheet to past and future climate change. *Nature* **608**, 275–286 (2022). https://doi.org/10.1038/s41586-022-04946-0

[7]  Ely, J.C., Clark, C.D., Hindmarsh, R.C.A., Hughes, A.L.C., Greenwood, S.L., Bradley, S.L., Gasson, E., Gregoire, L., Gandy, N., Stokes, C.R. and Small, D. (2021), Recent progress on combining geomorphological and geochronological data with ice sheet modelling, demonstrated using the last British–Irish Ice Sheet. J. Quaternary Sci., 36: 946-960. https://doi.org/10.1002/jqs.3098

[8]  Balco, G., Buchband, H., and Halberstadt, A. R. W.: 5 million year transient Antarctic ice sheet model run with "desensitized" marine ice margin instabilities, https://doi.org/https://doi.org/10.15784/601601, 2022a.
* * *
Jorjo Bernales, 01.01.2023.

---

## Author Response (AR1)

**Author response on 'Cosmogenic-nuclide data from Antarctic nunataks can constrain past ice sheet sensitivity to marine ice margin instabilities' by Halberstadt et al.**

Here we describe our revisions in response to comments by reviewers. The line numbers refer to the revised manuscript with tracked changes shown.

**Reviewer 1**

Figure 2: Box in panel (a) is not explained in caption; scale bar is missing; rainbow color scheme is unsuitable.
- The box has been removed, and figure has been corrected
- We have shifted to a colorblind-friendly palette

Figure 3: y-labels for panels (b,c) missing; time slices predate the model start time
- Axis labels have been corrected
- Labeling of the time slices has been corrected

Figure 4: It is unclear how the dashed black line for 'modern ice thickness' is established.
- Fig. 4 caption now refers the reader to Section 6.3 for further discussion on establishing a modern ice thickness

Figure 5: Caption does not describe the difference between the top and bottom scenario
- Caption now includes additional text ("Upper panels represent a hypothetical site with large differences between sensitized and desensitized models, and lower panels represent a hypothetical site with similar ice thickness behavior between models.")

Figure 7: Label for colorbar is missing; clarification needed regarding the blue dots
- Corrected, and clarification has been added in figure caption

Figures 9-13: Explain double y-axis, specify what 'h' is; add nuclide name to all figures consistently
- Figure captions are now consistent; 'h' is defined in the y-axis label, so it should now be clear (from discussion in Section 2) that 'Cumulative frequency of ice thickness' is an analogous quantity to 'Fraction of time ice-covered', the two y-axes.

Figure 14: Legend and caption are discrepant; white dots should be bigger
- Legend has been fixed to correctly match caption description; white dots adjusted

Clarify that different time periods of interest may exhibit different frequency behavior
- We have added text "...although the details of this frequency behaviour depend on the time period of interest." (L233)

The text states that the desensitized model is more likely to occupy elevations of ~1200m, but Fig. 4 looks like the blue curve spends relatively little time near 1200 m within the last 1-2 Ma

- 1200 m is the average model thickness, but we remove the reference to 1200m in the text to avoid confusion (L255)

Cite the data references for Figs 11 & 12 in main text
- Added (L439)

Consider using Jones et al. (2017) as a reference for the general approach described in this paper
- Added (L84, L95)

The reviewer also made several language suggestions and corrections, all of which have been corrected in the revised manuscript.

**Reviewer 2**

As described in our Response to Reviewer 2, we primarily address reviewer concerns through three groups of revisions. We also describe our revisions in response to additional specific reviewer concerns, listed below. In addition to these substantive revisions, the reviewer also made several language suggestions and corrections, all of which have been corrected in the revised version.

Revision 1: Clarify our aims and approach
- Additions in italics: "The aim of this paper is to explore how to use geologic data from the Antarctic continent to differentiate between ice sheet model simulations with *end-member instability behaviour* (e.g., Fig. 1b versus 1c). *We aim to elicit the largest possible variation in model ice sheet behaviour in order to test if this difference is resolvable using cosmogenic nuclide data.* We describe end-member simulations as 'sensitized' or 'desensitized' models based on the idea that stronger positive feedbacks in the form of marine ice instabilities result in model predictions that are more nonlinear, that is, more "sensitive," with respect to the forcing. Specifically, we investigate the sensitivity of ice sheets *to marine ice cliff instability under stronger and weaker ocean temperature forcing.*" (L42-50)
- Replaced sentence "At these locations, we can compare model predictions to geologic data as a means of gaining insight into whether sensitized or desensitized models are more accurate representations of ice sheet" with "At these locations, we can compare model predictions to geologic data as a means of gaining insight into *past ice sheet behaviour. This methodology therefore can be applied to future ensembles of simulations with more realistic and varied parametrizations to test which model realization most accurately represents the true ice sheet response to warm climate*" (L65-69)
- In abstract: Replace sentence "We identify regions of Antarctica where predicted frequency distributions are diagnostic of marine ice sheet instability parameterizations" with "We identify regions of Antarctica where predicted frequency distributions *reveal*

*differences in end-member ice-sheet behaviour*"

Revision 2: Properly describe our ocean forcing method as an external forcing rather than an internal nonlinearity
- ● Clarify that we have mixed our parameterization of nonlinear mechanisms with the amplitude of ocean forcing to elicit the largest possible model response:
    - ○ "Specifically, we investigate the sensitivity of ice sheets to (a) ocean temperatures and (b) *marine ice cliff instability mechanisms under stronger and weaker ocean temperature forcing*." (L48-50)
    - ○ "*To elicit the largest possible difference between these end-member simulations, we further enhance the ice sheet instability mechanisms in the sensitized model with a stronger ocean temperature forcing, whereas the desensitized model experiences weaker ocean forcing.*" (L148-150)
- ● Ensure that we no longer refer to our varied parameterizations as simply 'marine instabilities'.
    - ○ "The critical difference between sensitized ice sheet models (with strong marine ice margin instabilities *and strong ocean temperature forcing*) and desensitized models (with weak instabilities *and weak ocean forcing*) is the extent of deglaciation of marine basins." (L51-53)
    - ○ "This confirms that our model approach has successfully promoted 'linear' vs.'non-linear' ice-sheet behaviour by varying the parameterized ice sheet sensitivity to marine ice feedbacks *and ocean forcing*." (L239-241)
    - ○ The title is altered to: "Cosmogenic-nuclide data from Antarctic nunataks can constrain past ice sheet instabilities". We realize that this still invokes the presence of instabilities, but we feel that this title still accurately reflects our main point (which is that this methodology can be applied to model simulations to distinguish between instability parameterizations, and this is how we hope that the wider glaciological community will utilize this method in the future).

Revision 3: Provide more detail on the model components and application, input datasets, etc.
- ● We have added Appendix A (L546-579) to the revised manuscript that includes: (a) description of the long-term forcing scheme we applied here; (b) description of the ocean temperature ramped scaling; (c) explicit citations for input data fields; (d) description of the additional temperature shift added to the Amundsen and Bellingshausen Seas during interglacial time periods in our sensitized simulation; and (e) discussion of the insensitivity of results to model resolution
- ● This appendix is referenced in the main text on L191-192

Additional specific comments

Could you please elaborate on what you mean by "ambiguous"
- - We have removed the word for simplicity (L27)

Maybe a matter of opinion, but I would argue that is the present climate state the one

approaching the peak mPWP states, at least when acknowledging the poor paleorecord, uncertainties, and biases. Perhaps removing this sense of direction altogether?
- We have removed this comparison of modern and mid-Pliocene climate states (L28-29)

This would be a good place to introduce the concepts of ice shelf HF and MICI, after "… ice margins (Fig.1)" and before "Model runs…".
- Added sentence: "Specifically, [these non-linear feedback mechanisms] incorporate meltwater-driven hydrofracture of ice shelves, which can trigger full-thickness calving at the grounding line. Structural failure of exposed ice cliffs can drive rapid grounding-line retreat on a reverse-sloping bed, in a positive feedback loop dubbed 'marine ice cliff instability' (Section 3.1)" (L32-34)

[In places], MICI is mentioned, but not ice shelf HF. Consistency is needed on how you define and utilise these terms throughout the manuscript.
- After introducing these concepts, we refer to the combination of parameterizations (hydrofracturing and cliff retreat) as "marine ice cliff instability" (see L159, L172, L276)

Before such a hypothetical strong implication (L524 in original manuscript) could be trusted, a significant amount of validation would be needed.
- This sentence now reads: "If replicated at multiple sites across the continent, *and under a larger range of model experiments,* this would be an extremely significant result,…" (L540-541)

Figure 1: Needs a way to quantitatively interpret the contours and colormap, including the name of the plotted quantity. Adding the present-day observed grounding line and calving fronts in a distinctive manner, to allow for a quick evaluation of the model's ability to reproduce the present state of the ice sheet, would be useful to the reader as well.
- The caption has been updated to describe the contours (the colormap is redundant) as well as the plotted quantity: "Antarctic ice sheet model simulations with mid-Pliocene boundary conditions… showing ice surface elevation of grounded ice with contour lines at 500 m intervals and ice shelf boundaries as a thick grey line…" The figure has been altered so that subplot (a) shows modern grounding and calving line.

Figure 2: Needs stylistic consistency; black rectangle is unnecessary/inconsistent with panels; panels f and g are counterintuitive in that the linear response shows actually more retreat than the nonlinear one; needs a colorblind-friendly color palette
- Black rectangle is removed
- We now use a colorblind-friendly palette
- The panel with a non-linear (sensitized) response now shows greater advance and retreat than the linear (desensitized) panel

Figure 3: Time slices predate the model start time
- Labeling of the time slices has been corrected